# Ser/Thr kinase Trc controls neurite outgrowth in *Drosophila* by modulating microtubule-microtubule sliding

**Rosalind Norkett, Urko del Castillo, Wen Lu, Vladimir I Gelfand\***

Department of Cell and Developmental Biology, Feinberg School of Medicine, Northwestern University, Chicago, United States

**Abstract** Correct neuronal development requires tailored neurite outgrowth. Neurite outgrowth is driven in part by microtubule-sliding – the transport of microtubules along each other. We have recently demonstrated that a 'mitotic' kinesin-6 (Pavarotti in *Drosophila*) effectively inhibits microtubule-sliding and neurite outgrowth. However, mechanisms regulating Pavarotti itself in interphase cells and specifically in neurite outgrowth are unknown. Here, we use a combination of live imaging and biochemical methods to show that the inhibition of microtubule-sliding by Pavarotti is controlled by phosphorylation. We identify the Ser/Thr NDR kinase Tricornered (Trc) as a Pavarotti-dependent regulator of microtubule sliding in neurons. Further, we show that Trc-mediated phosphorylation of Pavarotti promotes its interaction with 14-3-3 proteins. Loss of 14-3-3 prevents Pavarotti from associating with microtubules. Thus, we propose a pathway by which microtubule-sliding can be up- or downregulated in neurons to control neurite outgrowth, and establish parallels between microtubule-sliding in mitosis and post-mitotic neurons.

## Introduction

In order to communicate, neurons must develop an extensive and precisely regulated network of axons and dendrites, collectively called neurites. Studying the mechanisms that form these processes is key to understanding early nervous system development. Neurites are filled with cytoskeletal components including microtubules. Neurons are exceptionally dependent on microtubules for long-range transport of cargo. Also, microtubule organization is essential for powering initial neurite outgrowth (*Kapitein and Hoogenraad, 2015*; *Lu et al., 2013*; *Winding et al., 2016*). In order to drive initial neurite outgrowth, microtubules themselves become the cargo and are transported relative to each other by molecular motors – a process known as microtubule sliding. Indeed, in cultured *Drosophila* neurons, microtubules can be seen pushing the plasma membrane at the tips of growing processes (*del Castillo et al., 2015*; *Lu et al., 2013*). Previous work from our group has identified the classical kinesin – Kinesin-1 – as the motor responsible for the majority of microtubule sliding in neurons (*Lu et al., 2013*; *Winding et al., 2016*).

Observation of microtubule sliding in neurons is of particular interest as this process is best described during vast cytoskeletal reorganization in mitosis, rather than in terminally differentiated neurons (*Baas, 1999*; *Del Castillo et al., 2019*). Microtubule sliding is observed in young neurons in culture, but decreases as neurons mature (*Lu et al., 2013*). Therefore, in addition to promoting neurite extension via microtubule sliding, there must also exist mechanisms to downregulate this process. This prevents overextension of neurites when their intended synaptic targets are correctly reached. Work from our group and others has previously identified the kinesin-6 Pavarotti/MKLP1 as a powerful regulator of microtubule-microtubule sliding. Depletion of Pavarotti/MKLP1 by RNAi leads to axon hyperextension and more motile microtubules (*Del Castillo et al., 2015*; *Lin et al., 2012*). Identifying a neuronal role for this kinesin was of interest as Kinesin-6 has well-studied roles

**\*For correspondence:**
vgelfand@northwestern.edu

**Competing interests:** The authors declare that no competing interests exist.

in mitosis. It exists as a heterotetramer with MgcRacGAP (Tumbleweed in *Drosophila*) to form the centralspindlin complex (*Adams et al., 1998*; *Basant and Glotzer, 2018*; *Mishima et al., 2002*). This complex bundles microtubules at the bipolar spindle in late anaphase (*Hutterer et al., 2009*). Here, it can locally activate RhoA and promote assembly of the contractile actin ring at the cortex, and so, cytokinesis (*Basant and Glotzer, 2018*; *Verma and Maresca, 2019*).

How Pavarotti itself is temporally regulated to inhibit microtubule sliding as neurons mature is unknown. Mitosis exhibits tight temporal regulation. We speculated that similar mechanisms might be at play in regulating Pavarotti activity in neurons with regard to microtubule sliding. One well-studied facet of centralspindlin regulation in regards to mitotic progression is that of phosphorylation (*Douglas et al., 2010*; *Guse et al., 2005*). Based on bioinformatics and a literature search, we targeted Ser/Thr kinases known to modify Pavarotti during mitosis and tested their ability to modulate microtubule sliding in interphase cells. One potential kinase was the NDR kinase Tricornered (Trc, LATS in mammals) – shown to phosphorylate MKLP1, the human ortholog of Pavarotti, in vitro at S710 (S745 in Pavarotti) (*Okamoto et al., 2015*; *Figure 1—figure supplement 1A*). Trc regulates cell cycle exit (*Hergovich et al., 2006*) and also has conserved roles in neurite outgrowth, described in *Drosophila*, *C. elegans* and mammals (*Emoto et al., 2006*; *Emoto et al., 2004*; *Gallegos and Bargmann, 2004*; *Ultanir et al., 2012*). How this kinase acts warrants further investigation.

Here, we use *Drosophila* S2 cells, neuronal culture and in vivo imaging to show Trc regulates microtubule sliding and dendrite outgrowth in neurons. We validate Pavarotti as a Trc substrate and demonstrate that phosphorylation of Pavarotti at S745 by Trc is necessary for proper control of microtubule sliding. We also show that phosphorylation of Pavarotti affects its subcellular distribution via interaction with 14-3-3 proteins in interphase cells – a mechanism conserved from mitosis. We demonstrate the function of this pathway in regulating development of *Drosophila* neurons.

## Results

### Tricornered kinase inhibits neurite outgrowth and microtubule sliding

We have previously demonstrated the requirement of microtubule-microtubule sliding, by kinesin-1, for neurite outgrowth in *Drosophila* (*Lu et al., 2013*; *Winding et al., 2016*). This sliding is opposed by the mitotic kinesin-6 'Pavarotti'/MKLP1 (*Del Castillo et al., 2015*). However, the mechanism by which Pavarotti itself is regulated in this neuronal context is unclear. We hypothesized that Pavarotti may be regulated by phosphorylation, as in mitosis (*Basant and Glotzer, 2017*; *Guse et al., 2005*; *Figure 1—figure supplement 1A*). We targeted kinases known to modify Pavarotti during mitosis (AuroraB, Plk1 and Trc), and tested their ability to modulate neuronal development and microtubule sliding in non-dividing cells. Initially, we measured microtubule sliding using the model system of S2 cells, a *Drosophila* cell line. We have previously demonstrated that kinesin-1 carries out microtubule-microtubule sliding in these cells and that Pavarotti inhibits this (*Del Castillo et al., 2015*; *Jolly et al., 2010*). Based on these preliminary experiments, we chose to focus on the NDR kinase Trc. To test the role of Trc, we decreased its levels using dsRNA. To measure microtubule sliding, we expressed a photoconvertible tubulin probe (tdEos-αTubulin84b) in S2 cells. Tubulin was photo-converted in a region of interest, and this specific population of microtubules was imaged by time-lapse confocal microscopy (*Barlan et al., 2013*). Sliding is measured as percentage of photoconverted tubulin outside the original photoconversion zone – see Materials and methods. We found a significant increase in microtubule sliding upon depletion of Trc compared to control (*Figure 1A*, quantified in B, *Video 1*). We also carried out sliding experiments in S2 cells to ensure this effect was specific to Trc depletion and not an off-target effect. To do this, we decreased Trc levels with dsRNA against a non-coding region of Trc, and co-expressed wild-type Trc. The non-coding dsRNA increased microtubule sliding and expression of wild-type Trc rescued this effect. Therefore, the effect is specific to Trc depletion (*Figure 2A,B*). As shown in *Figure 1—figure supplement 1 Figure 1B*, Trc is depleted and Pavarotti levels are unaffected.

Having established that Trc regulates microtubule sliding in S2 cells, we were next interested in investigating the potential role of Trc in neurite development. We analyzed primary neuronal cultures from third instar larvae expressing Trc RNAi. Neuron specific expression was achieved by elav gal4 >UAS Trc RNAi. This driver is derived from the elav locus which is expressed post-mitotically, exclusively in neurons and is active throughout development (*Robinow and White, 1988*). We found

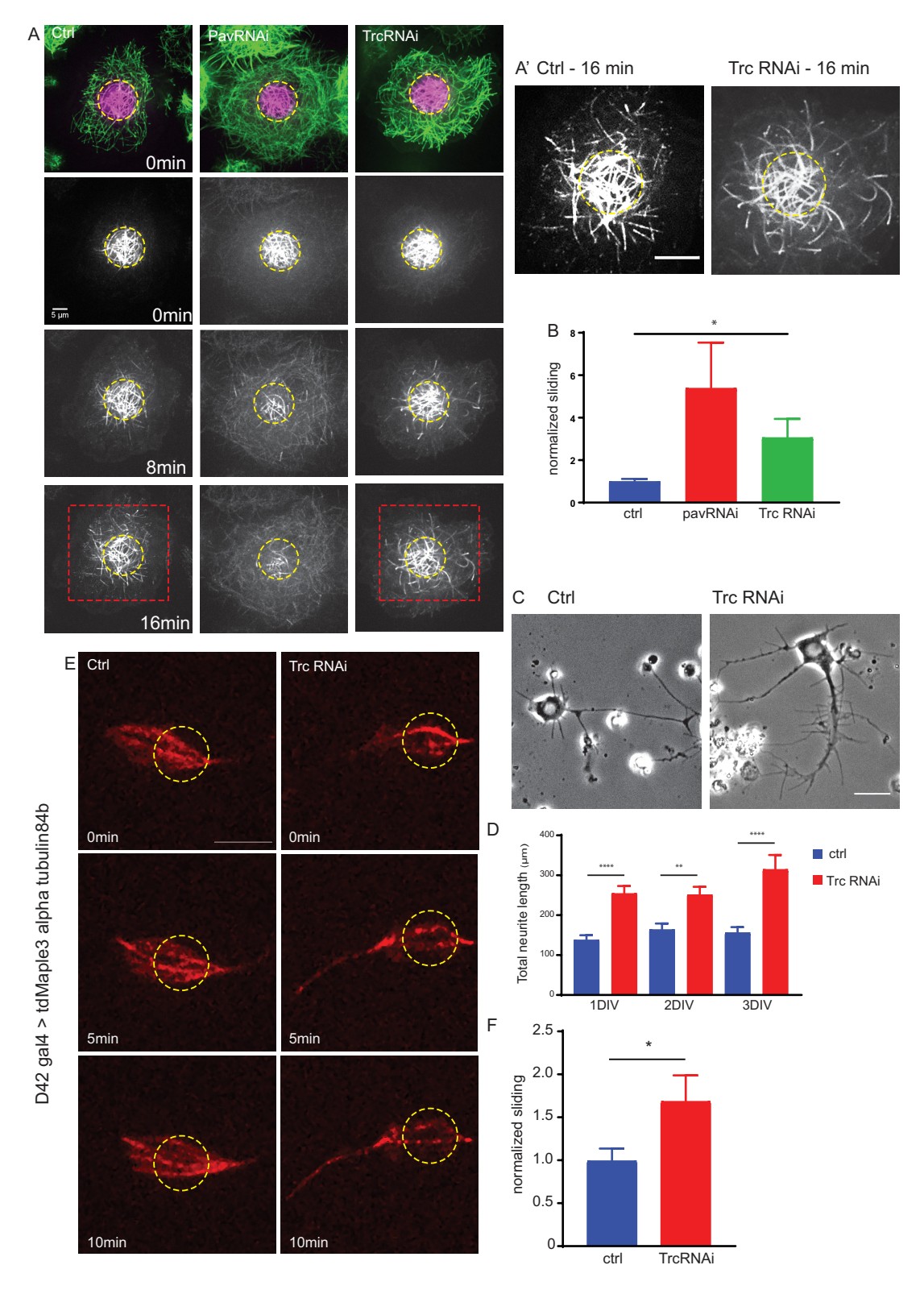

**Figure 1.** The kinase Trc regulates neurite outgrowth and microtubule sliding. (**A**) Example images of timelapse imaging to measure microtubule sliding in *Drosophila* S2 cells. Microtubules are shown at time zero in green. Magenta indicates photoconverted region. Photoconverted region is highlighted by dotted yellow line at other timepoints. Red box indicates region shown in inset in A' Scale bar 10 μm. (**B**) Quantification of microtubule sliding rate shows an increase upon Pavarotti or Trc depletion. n = 16–20 cells. Ctrl = 1.0 ± 0.1 upper 95% CI = 1.2, lower 95% CI = 0.8, Pav

*Figure 1 continued on next page*

*Figure 1 continued*

RNAi = 3.4 ± 1.0, 5.5, 1.4 Trc RNAi = 2.1 ± 0.4, 3.0, 1.2. Ctrl vs Trc RNAi p=0.03. (**C**) Representative images of 3rd instar larvae cultured neurons under control or elav > Trc RNAi conditions. (**D**) Quantification of total neurite length per cell over time in culture. The total neurite length is increased from control upon Trc depletion. 24 hr; ctrl = 137.8 ± 12.0 μm, Trc RNAi 254 ± 19.1 μm. p=0.0001. 48 hr; ctrl = 163.7 ± 15.3 μm, Trc RNAi = 251.2 ± 19.9 μm, p=0.001, 72 hr; ctrl = 156.3 ± 13.8 μm, TrcRNAi = 314.4 ± 36.5 μm, p=0.009. N = 11–23 cells from three independent experiments. (**E**) Example images from timelapse imaging of photoconverted microtubules in neurons under control conditions or upon Trc depletion. Tubulin was labeled with tdMaple3 alpha tubulin84b. After photoconversion, cells were imaged every minute for 10 min. Scale bar = 5 μm. (**F**) Quantification of microtubule sliding rates. Trc depletion leads to an increase in microtubule sliding rates in neurons. Ctrl = 1.0 ± 0.14, TrcRNAi = 1.69 ± 0.30. N = 20 cells from three independent experiments. p=0.04 Student's T-test.

The online version of this article includes the following source data and figure supplement(s) for figure 1:

**Source data 1.** Sliding rates for control, Pav RNAi and Trc RNAi-treated S2 cells.
**Source data 2.** Neurite length for neurons cultured from control larvae or Trc RNAi larvae.
**Source data 3.** Sliding rates for control, and Trc RNAi neurons.
**Figure supplement 1.** Domain structure of Pavarotti and demonstration of Trc knockdown.

a dramatic increase in total neurite length per cell in vitro (*Figure 1C and D*). Neuronal knockdown of Trc was confirmed by western blot of brain lysates of *Drosophila* 3rd instar larvae (*Figure 1—figure supplement 1C*).

Next, we directly tested the ability of Trc to regulate microtubule sliding in *Drosophila* cultured neurons. To do this, we expressed the photoconvertible tdMaple3 tubulin84b under the control of the motor neuron-specific D42 Gal4 driver and prepared dissociated neuronal cultures from brains of 3rd instar larvae (*Figure 1E*, *Video 2*). Photoconversion was carried out in a constrained region of the cell (as indicated by the yellow dotted line) and photoconverted signal was imaged over time to determine microtubule sliding rate, in a similar fashion to S2 cells. We compared control cells to those expressing Trc RNAi under the same driver. Consistent with our data in S2 cells, we found that depleting Trc levels led to an increase in microtubule sliding rate in cultured primary neurons (*Figure 1F*). Therefore, Trc has the ability to modulate microtubule sliding in order to control neurite outgrowth.

Together, these data describe a role for Trc as a negative regulator of neuronal development, independent from cell division, and suggest that the mechanism by which Trc regulates neurite outgrowth is via microtubule sliding. This effect could likely be intrinsic, rather than dependent on external cues, as the effect can be seen in dissociated cultures.

## Trc kinase activity is necessary to control microtubule sliding

We also confirmed that this effect on sliding was dependent upon the kinase activity of Trc in two ways. Firstly, we depleted the endogenous Trc with a dsRNA targeting a non-coding region and expressed either WT Trc, constitutively active Trc (Trc CA), or kinase-dead Trc (*Figure 2A,B*, *Video 3*; *He et al., 2005*). We confirmed expression of each of these mutants by fluorescence microscopy for BFP-Trc. We found that while WT and constitutively active Trc were able to reduce microtubule sliding to similar levels as control samples, the kinase dead mutant was not. Secondly, we depleted

Furry (Fry) in S2 cells. Furry is a large protein shown to promote Trc kinase activity without affecting expression level (*Figure 2—figure supplement 1A*; *Emoto et al., 2006*). Upon knockdown of Fry, and so decrease in Trc kinase activity, we found an increase in microtubule sliding (*Figure 2C,D Video 4*).

Further, we confirmed this effect in neurons by driving expression of photoconvertible maple tubulin, either alone or in conjunction with RNAi against Fry in motor neurons. We confirmed decrease of Fry protein level by immunostaining (*Figure 2—figure supplement 1B and C*). We carried out neuronal cultures from 3rd instar larvae and sliding experiments in the same way as

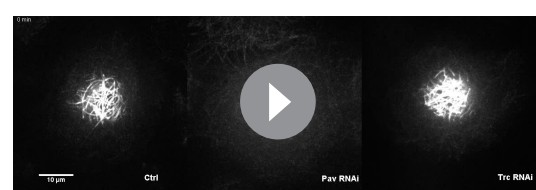

**Video 1.** Pavarotti and Trc RNAi increase microtubule sliding in S2 cells. Timelapse imaging of photoconverted microtubules in *Drosophila* S2 cells under control conditions or with depletion of Pavarotti or Trc. Microtubules were labeled with tdEOS alpha Tubulin 84b. 1 frame per minute.
https://elifesciences.org/articles/52009#video1

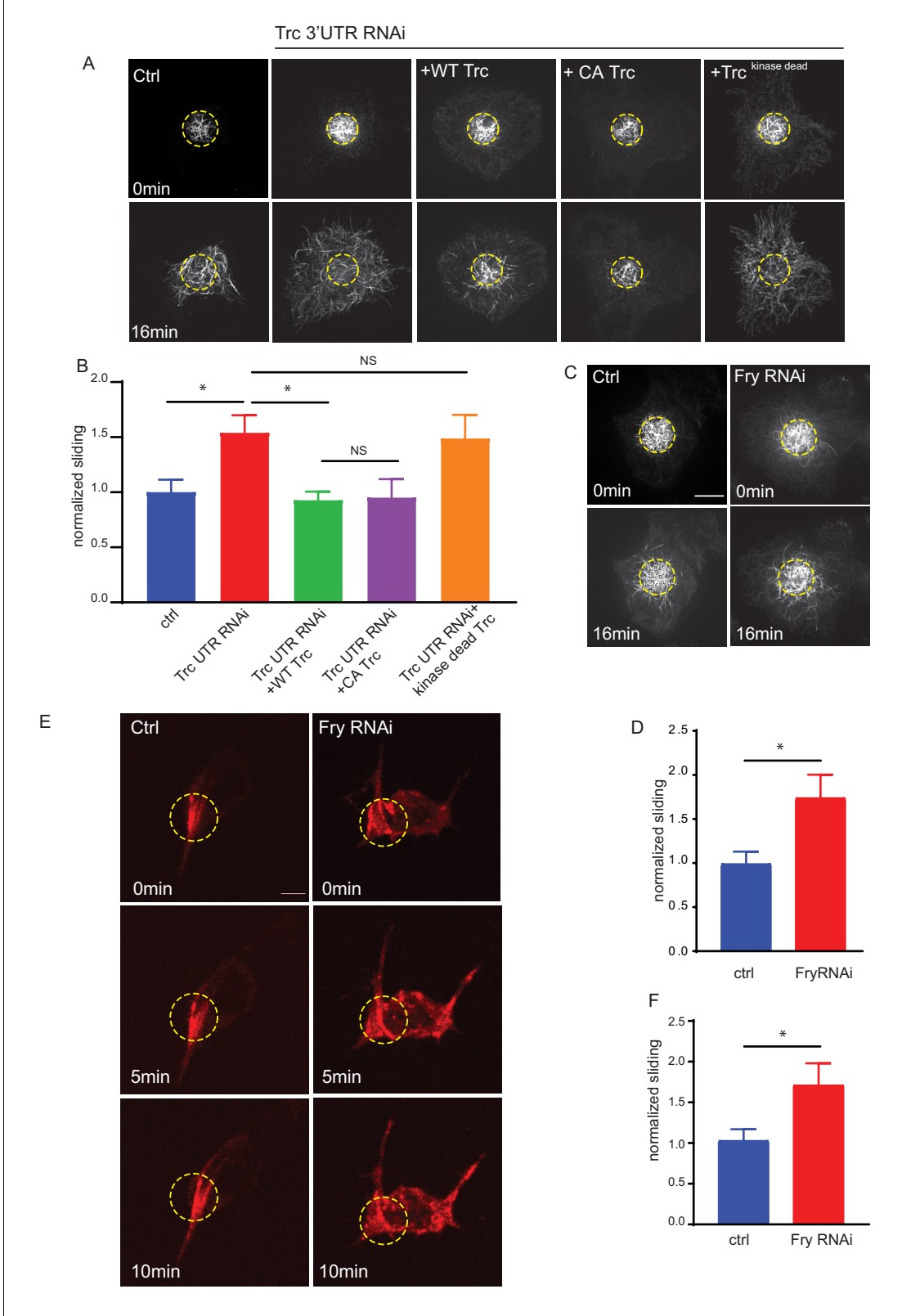

**Figure 2.** Trc regulates microtubule sliding in kinase-dependent manner. (**A**) Sliding experiments in S2 cells show increased sliding upon depletion of Trc with dsRNA targeting the non-coding region. The effect can be rescued with overexpression of WT Trc, but not kinase dead Trc. Scale bar 10μm. (**B**) Quantification of sliding experiments in A. n=18-50 cells from four independent experiments. Ctrl = 1.0 ± 0.11, Lower 95% CI 0.77, Upper 95% CI 1.23, Trc3'UTR RNAi = 1.54 ± 0.16, 1.218, 1.86, Trc3'UTR RNAi + WT Trc = 0.93 ± 0.08, 0.77, 1.09 Trc3'UTR RNAi + CA Trc = 0.95 ± 0.17, 0.59, 1.30,

*Figure 2 continued on next page*

*Figure 2 continued*

Trc3'UTR RNAi + kinase dead Trc = 1.49 ± 0.22, 1.05, 1.92. Ctrl vs Trc 3'UTR RNAi p = 0.03, Trc 3'UTR RNAi vs Trc 3'UTR RNAi + Trc WT p = 0.01. One-way ANOVA with Sidak's post hoc correction. (**C**) Sliding experiments in S2 cells show increased sliding upon depletion of Fry. Scale bar 10 μmD. Quantification of sliding experiments in C. n=26-37 cells from four independent experiments. Ctrl = 1 ± 0.13, Fry RNAi = 1.75 ± 0.26. Lower 95% CI ctrl = 0.72, Fry RNAi 1.23. Upper 95% CI ctrl = 1.268, Fry RNAi = 2.265. p = 0.024 Student's T-test. (**E**) Representative images from timelapse imaging of photoconverted microtubules in neurons under control conditions or upon Fry depletion. Tubulin was labeled with tdMaple3 alpha tubulin84b. After photoconversion, cells were imaged every 30s for 10 min. Scale bar = 5μm (**F**). Quantification of microtubule sliding rates. Fry depletion leads to an increase in microtubule sliding rates in neurons. n = 31-42 cells from four independent experiments. Ctrl = 1.0 ± 0.13, 0.78, 1.3, Fry RNAi = 1.7 ± 0.26, 2.3, 1.2. Ctrl vs Fry RNAi p = 0.02 Student's T-test.

The online version of this article includes the following source data and figure supplement(s) for figure 2:

**Source data 1.** Microtubule sliding rates in S2 cells for Trc knockdown and rescue experiments.
**Source data 2.** Microtubule sliding rates in S2 cells for control and Fry knockdown S2 cells.
**Source data 3.** Sliding rates for control and Fry RNAi neurons.
**Figure supplement 1.** Confirming Fry knockdown.
**Figure supplement 1—source data 1.** Fry immunostaining intensities.

for Trc knockdown. We found that depleting Fry, and so decreasing Trc kinase activity, led to an increase in microtubule sliding (*Figure 2E,F Video 5*) Taken together, these experiments show that the kinase activity of Trc is required to regulate microtubule sliding and that these findings are consistent between *Drosophila* neurons and S2 cells.

## Tricornered kinase phosphorylates Pavarotti to Brake microtubule sliding

Next, we investigated if the effect of Trc on microtubule sliding was dependent upon Pavarotti. In order to address this, we carried out sliding assays in S2 cells. We overexpressed Trc, either alone, or in conjunction with Pavarotti knockdown. Overexpression of Trc in S2 cells resulted in a decrease in microtubule sliding. This is in good agreement with our data describing Trc as a negative regulator of sliding (*Figure 3A,B*, *Video 6*). Importantly, upon depletion of Pavarotti, this decrease in sliding was lost (*Figure 3A,B*). Therefore, Pavarotti must be present for Trc to oppose microtubule-microtubule sliding.

In order to confirm Pavarotti was indeed phosphorylated by Trc at the predicted site S745, we carried out immunoblotting with a phospho-specific antibody for Pavarotti S745. We expressed GFP Pavarotti in HEK 293FT cells and performed pull downs with anti-GFP antibody. Western blot analysis of precipitated GFP Pavarotti showed basal phosphorylation at S745. Mutation of Pavarotti Ser745 to Ala eliminated signal with the phospho-specific antibody, confirming the antibody specificity. Importantly, ectopic expression of constitutively active Trc (Trc CA), resulted in a roughly twofold increase in Pavarotti phosphorylated at S745. Therefore, Trc phosphorylates Pavarotti at Ser 745 in cells (*Figure 3I*). Notably, we confirmed these findings in brain tissue (*Figure 3J*). We dissected brains from either control 3rd instar larvae or brains of those expressing Trc RNAi under control of the pan neuronal driver elav Gal4. We immunoprecipitated endogenous Pavarotti from these brains and immunoblotted for phosphorylation of Pavarotti at serine 745. We found that phospho Pavarotti was reduced by around 35% upon Trc depletion when compared to total immunoprecipitated Pavarotti. Therefore, Trc phosphorylates Pavarotti at serine 745 in neurons in vivo and depletion of the kinase leads to a greater pool of unphosphorylated Pavarotti at S745.

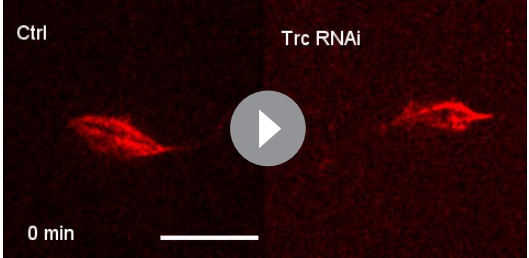

**Video 2.** Trc depletion increases microtubule sliding in neurons. Timelapse imaging of photoconverted microtubules in *Drosophila* cultured neurons under control conditions or with depletion of Trc. Microtubules were labeled with UAS tdMaple alpha Tubulin 84b under control of a motor neuron specific D42 gal4 driver. Trc was depleted with UAS Trc RNAi under control of a motor neuron-specific D42 gal4 driver. 1 frame per minute. Scale bar 10 μm.
https://elifesciences.org/articles/52009#video2

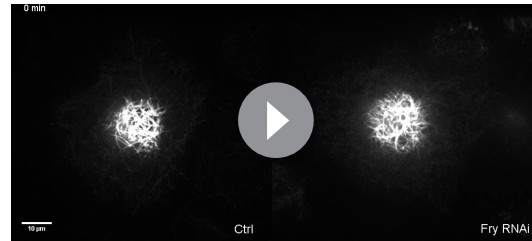

**Video 3.** Microtubule sliding in S2 cells is dependent upon Trc kinase activity. Timelapse imaging of photoconverted microtubules in *Drosophila* S2 cells under control conditions or with depletion of Trc using dsRNA against a non-coding region. Expression of WT Trc can rescue the effect of Trc RNAi. Expression of Kinase dead Trc is unable to rescue the effect of Trc RNAi. Microtubules were labeled with tdEOS alpha Tubulin 84b. 1 frame per minute.
https://elifesciences.org/articles/52009#video3

**Video 4.** Fry depletion increases microtubule sliding in S2 cells. Timelapse imaging of photoconverted microtubules in *Drosophila* S2 cells under control conditions or with depletion of Furry (Fry). Microtubules were labeled with tdEOS alpha Tubulin 84b. 1 frame per minute.
https://elifesciences.org/articles/52009#video4

Next, we tested if phosphorylation at S745 was necessary for Pavarotti to inhibit microtubule sliding. We once more carried out sliding assays, this time with a knockdown and rescue approach. Knockdown of Pavarotti by this approach was confirmed by western blot. Expression of WT and S745A GFP Pavarotti was confirmed by immunofluorescence (*Figure 3—figure supplement 1*). Depletion of Pavarotti with a dsRNA against a non-coding region increased sliding which could be reduced again by expression of wild-type Pavarotti. However, expression of Pavarotti S745A mutant failed to reduce sliding levels to their control levels (*Figure 3C,D*, *Video 7*). Therefore, Pavarotti can no longer inhibit microtubule sliding when phosphorylation of S745 is prevented.

To understand why phosphorylation of Pavarotti is required for sliding inhibition, we compared microtubule binding of GFP Pavarotti in S2 cells under control conditions, after Trc depletion, or after mutation of S745. In order to visualize microtubule-bound Pavarotti, we extracted S2 cells with Triton X-100 under conditions preserving microtubules (see Materials and methods). This approach removes soluble Pavarotti from the cell. Therefore, we analyzed microtubule area that colocalized with Pavarotti signal. In this way, nuclear Pavarotti is excluded from the analysis. A more detailed description of this analysis and images in individual channels are shown in *Figure 3—figure supplement 2A and B*. We found that depletion of Trc led to around a 50% decrease in association of Pavarotti with microtubules (*Figure 3E,F*). Similar experiments with the phospho-null mutant S745A also show a decrease in association of Pavarotti with microtubules (*Figure 3G,H*). Therefore, Pavarotti in its unphosphorylated state has lower affinity for microtubules. Under these conditions, microtubule sliding is permitted.

## Inhibition of microtubule sliding by pavarotti requires interaction with 14-3-3 proteins

Our data so far show a role for phospho-regulation of Pavarotti in microtubule sliding, beyond its canonical function in cytokinesis. We chose to continue investigating parallels between mitosis and interphase/post-mitotic microtubule sliding, this time testing the involvement of 14-3-3 proteins. These proteins have been shown to form a complex with Pavarotti dependent upon phosphorylation at the identified S745 site (*Douglas et al., 2010*; *Fesquet et al., 2015*). This association influences microtubule bundling (*Douglas et al., 2010*). We chose to further probe this mechanism, both with regard to Pavarotti S745 phosphorylation by Trc and

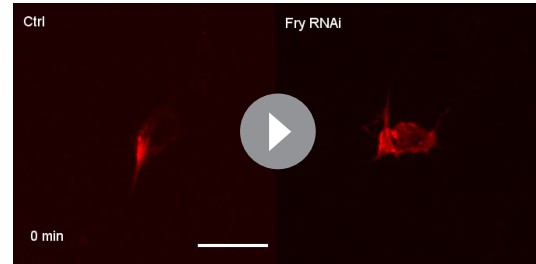

**Video 5.** Fry depletion increases microtubule sliding in neurons. Timelapse imaging of photoconverted microtubules in *Drosophila* cultured neurons under control conditions or with depletion Fry. Microtubules were labeled with UAS tdMaple alpha Tubulin 84b under control of a motor neuron specific D42 gal4 driver. Fry was depleted with UAS Fry RNAi under control of a motor neuron specific D42 gal4 driver. 2 frames per minute. Scale bar 10 μm.
https://elifesciences.org/articles/52009#video5

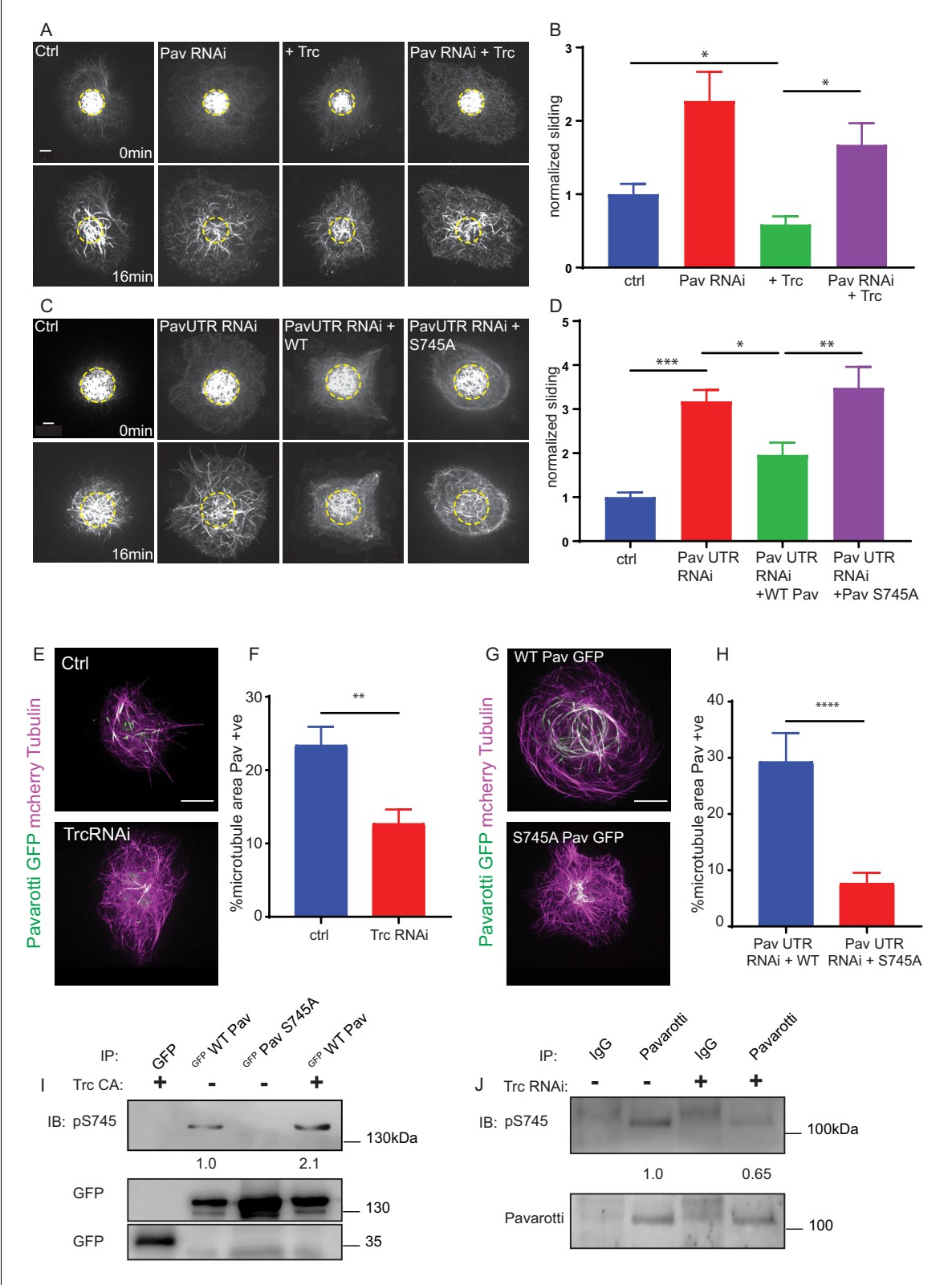

**Figure 3.** Trc regulates microtubule sliding via phosphorylation of Pavarotti. (**A**) Sliding experiments in S2 cells show a decrease in microtubule sliding with Trc overexpression. Trc overexpression in conjunction with depletion of Pavarotti increases sliding beyond control levels Scale bar 5 μm. (**B**) Quantification of sliding experiments in A. n = 27–41 cells per condition. Ctrl = 1.0 ± 0.11, Upper 95% CI = 1.2, Lower 95% CI = 0.8, Pav RNAi = 1.6 ± 0.25, 2.1, 1.0, Trc OE = 0.61 ± 0.0.8, 0.8, 0.44, Pav RNAi + Trc OE = 1.4 ± 0.22, 1.9, 0.9. Ctrl vs Trc OE p=0.04, Trc OE vs Pav RNAi + Trc

*Figure 3 continued on next page*

*Figure 3 continued*

OE p=0.003. One-way ANOVA with Sidak's post-hoc test. (**C**) Sliding experiments in S2 cells show an increase in microtubule sliding with Pavarotti depletion. The effect can be rescued with WT Pavarotti but not with Phospho null mutant S745A. (**D**) Quantification of sliding experiments shown in C. n = 39–48 cells from 4 independent experiments. Ctrl = 1 ± 0.10 Upper 95% CI = 1.2, Lower 95% CI = 0.79, Pav RNAi = 3.16 ± 0.26, 2.65, 3.70, Pav RNAi + WT = 1.96 ± 0.28, 1.40, 2.52 Pav RNAi + Pav S745A = 3.49 ± 0.48, 2.53, 4.44. Ctrl vs Pav RNAi p=0.0001, Pav RNAi vs pav RNAi + WT p=0.04, Pav RNAi vs Pav RNAi + S745A p=0.94, Pav RNAi + WT vs Pav RNAi + S745A p=0.0025. One-way ANOVA with Sidak's post hoc correction. (**E**) Example images of extracted S2 cells expressing mCherry Tubulin and WT Pavarotti GFP under control or Trc RNAi conditions. (**F**) Quantification of microtubule area colocalized with Pavarotti. n = 18–26 cells from three independent experiments. Ctrl = 23.5 ± 2.4%, Upper 95% CI = 28.6, Lower 95% CI = 18.3 Trc RNAi = 12.78 ± 1.9%, 16.62, 8.94. p=0.001 Student's T-test Scale bar = 10 μm. (**G**) Example images of extracted S2 cells expressing mCherry Tubulin and WT Pavarotti GFP or Pavarotti S745A GFP. Endogenous Pavarotti was depleted with dsRNA targeting non-coding regions. (**H**) Quantification of microtubule area colocalized with Pavarotti. n = 12–17 cells from three independent experiments. Scale bar = 10 μm. WT = 29.4 ± 5.0%, Upper 95% CI = 40.4, Lower 95% CI = 18.4, S745A = 7.79 ± 1.8%, 11.5, 4.08. p=0.0001 Student's T-test (**I**) Western blot of S745 phospho-Pavarotti from HEK cell lysates shows an increase in this species with Trc overexpression. (**J**) Western blot of immunoprecipitated Pavarotti prepared from dissected 3rd instar larvae brains shows a decrease in phospho Pavarotti at S745 upon depletion of Trc.

The online version of this article includes the following source data and figure supplement(s) for figure 3:

**Source data 1.** Microtubule sliding rates for control, Pav knockdown and Trc overexpressing cells.
**Source data 2.** Microtubule sliding rates for Pavarotti knockdown and rescue experiments.
**Source data 3.** Pavarotti localization in control and Trc RNAi-treated cells.
**Source data 4.** Pavarotti WT and S745A localization.
**Figure supplement 1.** Pavarotti knockdown and rescue.
**Figure supplement 1—source data 1.** Pavarotti WT and S745A expression levels.
**Figure supplement 2.** Method for determining Pav-positive microtubule area.

microtubule sliding. Initially, we carried out co-immunoprecipitation experiments from HEK293 FT cells. We over- expressed WT or S745A GFP Pavarotti. We found a robust interaction between WT Pavarotti and endogenous 14-3-3 ξ. Mutation of S745 to alanine, mimicking a non-phosphorylated form of Pavarotti, abrogated the interaction. Co-expression of exogenous Trc with GFP Pavarotti, generating a greater pool of phosphorylated Pavarotti at S745, increases the interaction between Pavarotti and 14-3-3 ξ by roughly 60% (*Figure 4A*). Therefore, we confirm that phosphorylation of Pavarotti, by Trc, leads to the recruitment of 14-3-3 proteins.

We next tested if the interaction between Pavarotti and 14-3-3s is necessary for microtubule sliding inhibition by Pavarotti in S2 cells. We depleted *Drosophila* 14-3-3 β and ξ isoforms in S2 cells by dsRNA and overexpressed Pavarotti. Knockdown is demonstrated by western blot in *Figure 4F*. Overexpression of Pavarotti caused a decrease in microtubule sliding. However, when we depleted 14-3-3 levels, we no longer observed this decrease in sliding upon Pavarotti overexpression (*Figure 4B*, quantified in C, *Video 8*). Thus, Pavarotti is not capable of inhibiting microtubule sliding in the absence of 14-3-3 proteins. These data are in good agreement with microtubule sliding assays performed with our S745A mutant (*Figure 3D*), where a complex between Pavarotti and 14-3-3s does not form. Observation of the microtubule network in this condition showed a decrease in GFP Pavarotti associated with microtubules (*Figure 4D,E*, analysis carried out as in *Figure 3—figure supplement 2A*, individual channels presented in *Figure 3—figure supplement 2B*), consistent with the effect seen with Trc depletion or the S745A mutation. Altogether, our data suggest Pavarotti locally brakes microtubule sliding and this is facilitated by interaction with 14-3-3 proteins. The formation of this complex requires phosphorylation at S745, by the kinase Trc.

## Pavarotti and Trc act in the same pathway to control microtubule sliding in neurons and dendrite outgrowth in vivo

Based on our data in S2 cells and *Drosophila* brain tissue, we were curious to determine if our proposed mechanism for microtubule sliding was replicated in neurons. Data in *Figures 1F*

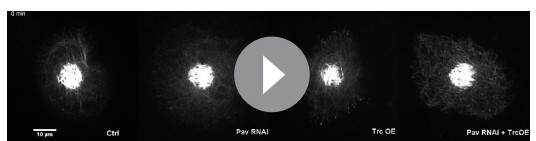

**Video 6.** Trc requires Pavarotti to inhibit microtubule sliding in S2 cells. Timelapse imaging of photoconverted microtubules in *Drosophila* S2 cells. Trc overexpression inhibits microtubule sliding only in the presence of Pavarotti. Microtubules were labeled with tdEOS alpha Tubulin 84b. 1 frame per minute.
https://elifesciences.org/articles/52009#video6

**Video 7.** Pavarotti phosphorylation at Ser745 is required to inhibit sliding. Timelapse imaging of photoconverted microtubules in *Drosophila* S2 cells under control conditions or with depletion of Pavarotti using dsRNA against a non-coding region. Expression of WT Pav can rescue the effect of Pav RNAi. Expression of Phopsho-null Pavarotti is unable to rescue the effect of Pav RNAi. Microtubules were labeled with tdEOS alpha Tubulin 84b. 1 frame per minute.
https://elifesciences.org/articles/52009#video7

and *3J* show that Trc regulates sliding in neurons and phosphorylates Pavarotti in neurons. To test if these observations were part of the same pathway, we carried out microtubule sliding assays in cultured neurons from *Drosophila* 3rd instar larvae upon depletion of Pavarotti, Trc, or both in concert. We found that, like Trc, depleting Pavarotti led to a roughly twofold increase in microtubule sliding rate in neurons (*Figure 5A* quantified in B, *Video 9*). This is consistent with our previous work showing the same effect in *Drosophila* embryonic neurons (*Del Castillo et al., 2015*). Importantly, knockdown of both Pavarotti and Trc together increased microtubule sliding to similar levels to those of either knockdown individually – double knockdown had no additive effect. This supports the hypothesis that Pavarotti and Trc regulate microtubule slid-

ing in neurons via the same pathway decreasing Trc levels in neurons led to a decrease in phospho-Pavarotti, and it is the lack of this species that prevents the brake to microtubule sliding.

We hypothesized that Trc and Pavarotti are in the same pathway in *Drosophila* neurons in vivo and that their coordination of microtubule sliding regulates neurite outgrowth. In order to test this, we measured the dendritic arbor of class IV DA (dendritic arborization) neurons (sensory neurons) in 3rd instar larvae either upon knockdown of Trc, Pavarotti, or both in conjunction. We have previously demonstrated that dendrite outgrowth in these cells is dependent upon microtubule sliding as expression of a sliding-deficient kinesin-1 leads to shorter dendritic arbors (*Winding et al., 2016*). Class IV neurons were labeled with ppk::tdTomato (membrane localized tdTomato under direct control of the classIV DA neuron specific pickpocket promoter) and pan-neuron specific expression of RNAis was achieved with elav gal4, which is expressed in all peripheral neurons, including the class IV DA neurons, (*Luo et al., 1994*). Knockdown of Pavarotti and Trc together was confirmed by western blot as presented in *Figure 5—figure supplement 1A*. In each case, we observed a roughly 40% increase in total dendrite length (*Figure 5C,D*). In the case of Trc, these data are consistent with previous reports (*Emoto et al., 2004*) where expression was abolished by mutation rather than neuron-specific RNAi. Notably, the effect upon double knockdown was equivalent to that of the single knockdowns (*Figure 5C,D*). This supports the hypothesis that Trc and Pavarotti regulate neurite outgrowth in concert.

As well as total dendritic length, we also calculated the number of branch points per cell (*Figure 5—figure supplement 1B*). Trc has been previously shown to increase dendrite branching (*Emoto et al., 2004*). Our analyses reproduced these findings – Trc RNAi cells had roughly 30% more branch points than controls. Interestingly, Pavarotti depletion had no effect on number of branch points, and depletion of Trc and Pavarotti together mimics the effect of Trc RNAi alone. From these data, we infer that Trc and Pavarotti regulate neurite outgrowth together, but that Trc has another role, independent of Pavarotti phosphorylation, in regulating dendritic branching.

Taken together, our data demonstrate that Trc regulates microtubule sliding in neurons together with Pavarotti and that this kinase phosphorylates Pavarotti at S745. Pavarotti phosphorylation at S745 is required to brake microtubule sliding and so, correctly tailor neurite extension.

## Discussion

Developing neurons must extend neurites to form a network for correct communication. This outgrowth must be downregulated as the neurites reach their intended targets and form stable synapses. We have previously shown that microtubule-microtubule sliding is required for neurite outgrowth in young neurons and is diminished in mature neurons by the action of *Drosophila* kinesin-6, Pavarotti. However, the processes by which Pavarotti might temporally regulate microtubule sliding were unknown. Here, we report an inhibitory pathway for microtubule sliding. Our previous work has demonstrated roles for 'mitotic' processes (microtubule-microtubule sliding) and 'mitotic'

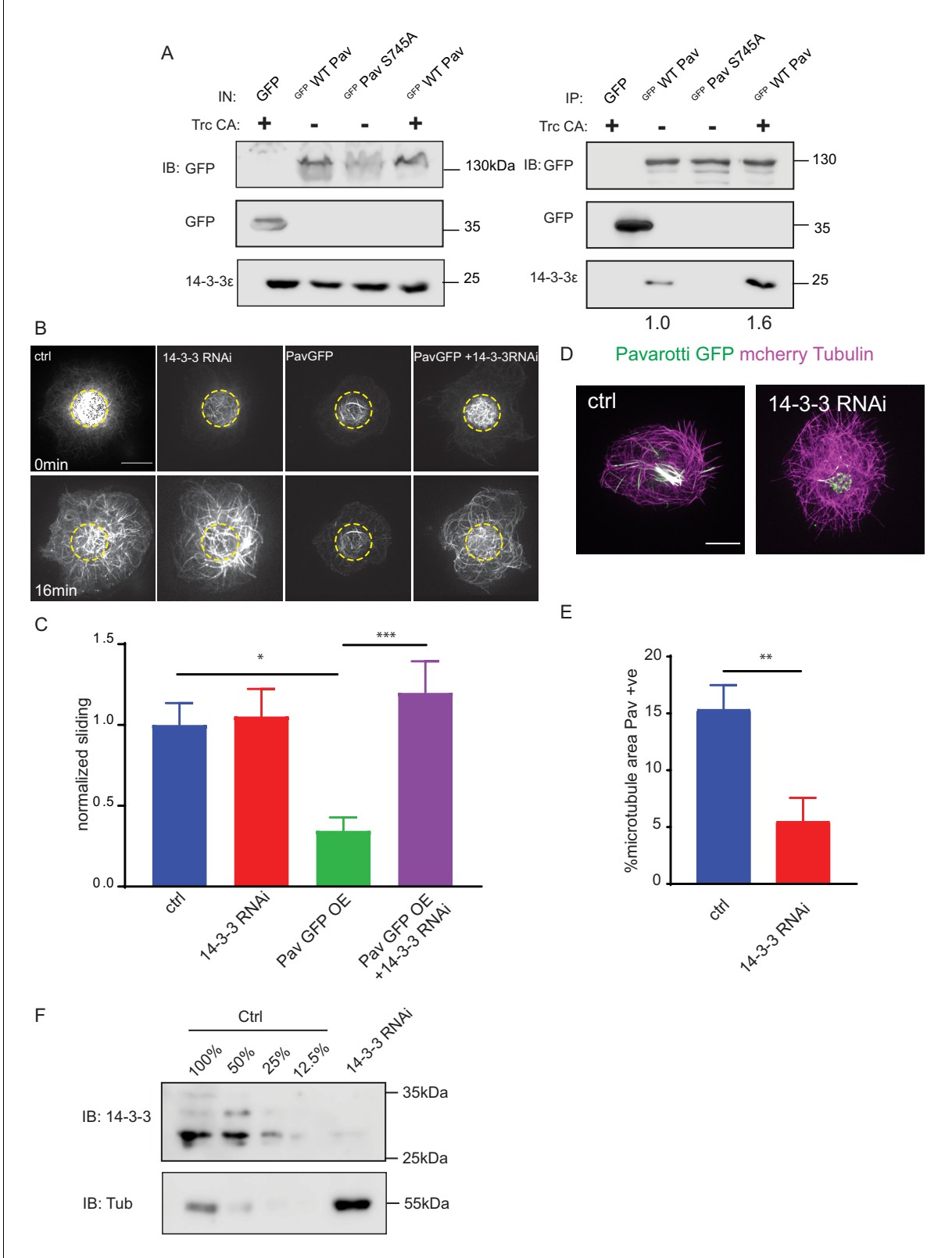

**Figure 4.** Phospho-Pavarotti brakes microtubule sliding via interaction with 14-3-3 proteins. (**A**) Western blot from HEK cell lysate showing co immunoprecipitation of Pavarotti and 14-3-3. The interaction is lost upon mutation of S745 to Alanine and increased upon co-expression of the kinase Trc. (**B**) Sliding experiments in S2 cells show the ability of Pavarotti to brake microtubule sliding is dependent upon 14-3-3 proteins. Scale bar = 10µm. (**C**) Quantification of sliding experiments in B. n=21-26 cells from three independent experiments. Ctrl = 1 ± 0.14 Upper 95% CI = 1.28, Lower

*Figure 4 continued on next page*

Figure 4 continued

95% CI = 0.72, PavOE = 0.35 ± 0.08, 0.52, 0.18 14-3-3 RNAi = 1.05 ± 0.17, 1.41, 0.70, 14-3-3RNAi + PavOE = 1.20 ± 0.20, 1.6, 0.8. ctrl vs pav OE p = 0.01, ctrl vs 1433RNAi p = 0.99, ctrl vs 1433 RNAi + PavOE p = 0.78, Pav OE vs 1433 RNAi p = 0.007, pav OE vs 1433 RNAi + Pav OE p = 0.0004, 1433 RNAi vs 1433 RNAi + Pav OE p = 0.91 One-way ANOVA with Tukey's post-hoc correction. (D) Example images of extracted S2 cells expressing mCherry Tubulin and WT Pavarotti GFP. Depletion of 14-3-3s decreases microtubule area decorated with Pavarotti. Scale bar = 10μm (E). Quantification of microtubule area colocalized with Pavarotti. n = 22-26 cells from three independent experiments. Ctrl = 15.4 ± 2.1, Upper 95% CI = 19.72, Lower 95% CI = 11.07, 14-3-3 RNAi = 5.54 ± 2.0, 9.78, 1.30. p = 0.0017 Student's T-test. (F) Western blot from S2 cell lysate demonstrating knockdown of 14-3-3. The online version of this article includes the following source data and figure supplement(s) for figure 4:

Source data 1. Sliding rates for 14-3-3 RNAi and Pavarotti overexpression experiments.
Source data 2. Pavarotti localization with 14-3-3 RNAi.
Figure supplement 1. Confirming GFP Pavarotti expression.

motors (kinesin-6 Pavarotti) in regulating the neuronal cytoskeleton. In this study, we extended these parallels to identify the NDR kinase Trc (Tricornered) as a required component of the Pavarotti pathway, regulating neurite outgrowth (*Figure 6*).

## Trc is a novel regulator of microtubule sliding

NDR kinases have well-studied roles in cell division and tissue morphogenesis. The yeast homologue of Trc (Dbf2p) promotes chromosome segregation and mitotic exit. These functions are conserved in mammals (*Hergovich et al., 2006*; *Tamaskovic et al., 2003*). However, neuronal expression of some NDR kinases has additionally been reported. In neurons, depletion of Trc has been linked to increased outgrowth of both axons and dendrites across multiple taxa (*Emoto et al., 2004*; *Gallegos and Bargmann, 2004*; *Ultanir et al., 2012*; *Zallen et al., 1999*). Our data are in good agreement with these previous reports as we measure an increase in neurite length in vitro and an increase in dendrite length in vivo. Further, our data uncover a mechanism for this increased outgrowth. Depletion of Trc leads to increased microtubule sliding in both S2 cells and in cultured primary neurons. This increased sliding allows microtubules to push at the tips of nascent neurites, providing the force required for their extension (*del Castillo et al., 2015*; *Lu et al., 2013*). Indeed, this sliding has been shown to translate into dendrite outgrowth in vivo – expression of a sliding deficient kinesin-1 mutant drastically decreases the dendritic arbors of *Drosophila* sensory neurons (Class IV DA neurons) (*Winding et al., 2016*).

## Trc phosphorylates Pavarotti to Brake microtubule sliding

Similarly to Trc, the kinesin-6 Pavarotti was thoroughly studied with regard to cell division. It is a microtubule cross linker and signaling hub to promote cleavage furrow ingress (*Adams et al., 1998*; *Basant and Glotzer, 2017*; *Verma and Maresca, 2019*). Moreover, Pavarotti's ability to localize to the spindle is dependent on its phosphorylation state (*Guse et al., 2005*). Here, we have shown that Pavarotti is a downstream effector of Trc in the sliding inhibition pathway – Trc overexpression could only decrease sliding in the presence of Pavarotti. In agreement with these data, in neurons, depletion of both Pavarotti and Trc together had the same effect as depletion of either protein individually. Using a similar approach, we have also demonstrated that Trc's ability to regulate microtubule sliding is dependent upon its kinase activity. Knockdown and rescue experiments in S2 cells showed wild type and constitutively active Trc constructs could restore normal sliding levels but, a kinase dead variant was unable to do this. This is supported by data in neurons showing that decrease of Fry protein level increases microtubule sliding. Decreasing Fry decreases Trc kinase activity without affecting protein level (*Emoto et al., 2004*).

Further we have used a phospho-null mutant to demonstrate that phosphorylation of Pavarotti at the proposed Trc site of Serine 745 is necessary to inhibit sliding. Importantly, we confirm

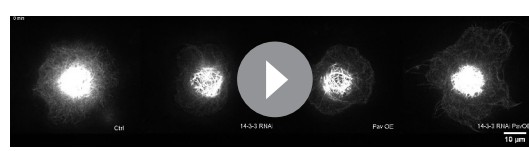

**Video 8.** 14-3-3 proteins are required for Pavarotti to inhibit microtubule sliding. Timelapse imaging of photoconverted microtubules in *Drosophila* S2 cells. Pavarotti can inhibit sliding only in the presence of 14-3-3 proteins. Microtubules were labeled with tdEOS alpha Tubulin 84b. 1 frame per minute.
https://elifesciences.org/articles/52009#video8

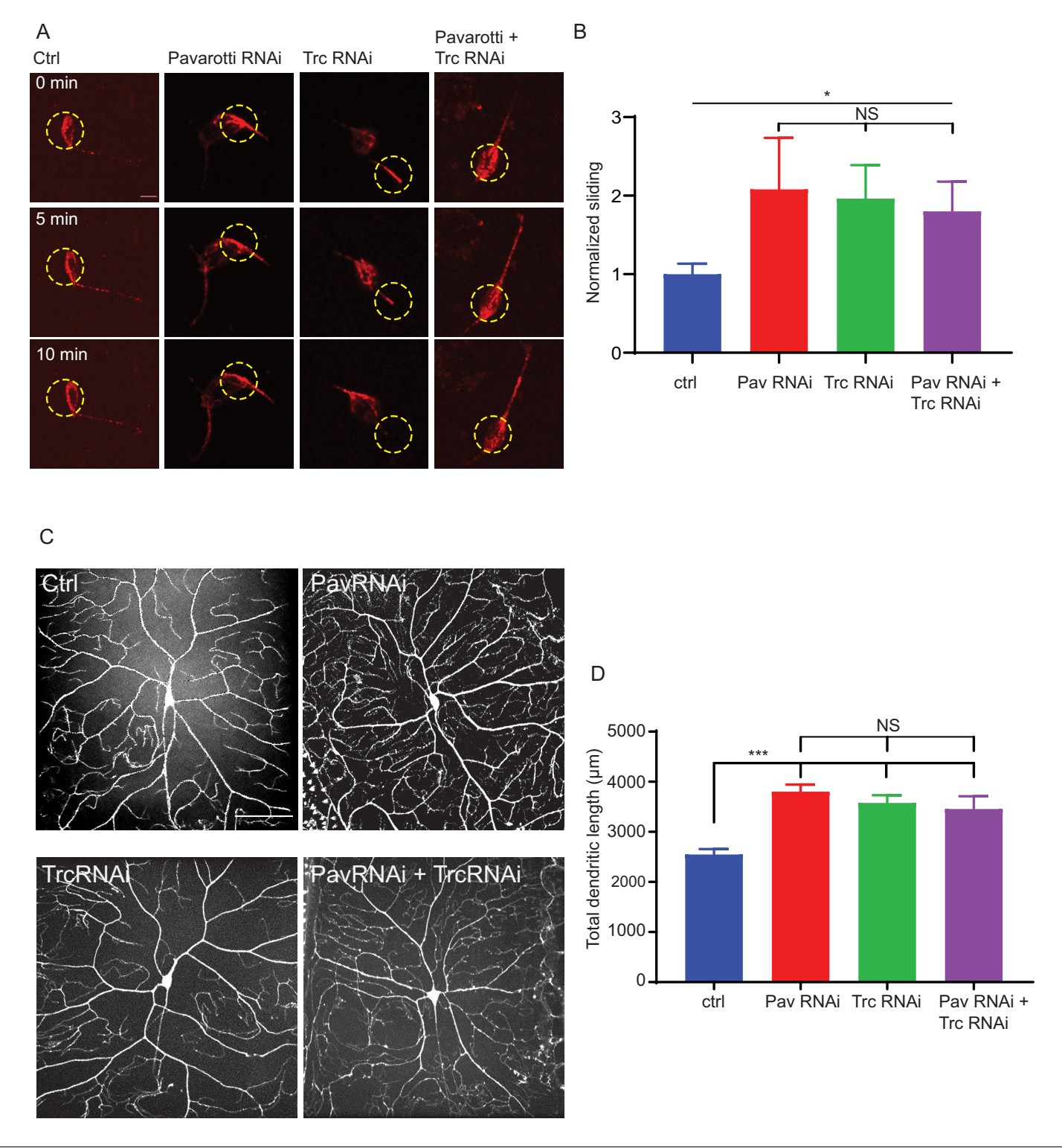

**Figure 5.** Pavarotti and Trc act in the same pathway to control dendrite outgrowth in vivo. (**A**) Example images from timelapse imaging of photoconverted microtubules in neurons under control conditions or upon Pavarotti, Trc or Pavarotti and Trc depletion. Scale bar = 10 µm (**B**) Quantification of microtubule sliding rates. Ctrl = 1.0 ± 0.1, Upper 95% CI = 1.8, Lower 95% CI = 0.7, Pav RNAi = 2.1 ± 0.7, 3.5, 0.6, Trc RNAi = 2.0 ± 0.4, 2.8, 1.1, Pav and Trc RNAi = 1.8 ± 0.4, 2.6, 1.0. Ctrl vs Pav and Trc RNAi p = 0.014 Student's T-test. (**C**) Example images showing DA neurons labeled with ppk::tdTomato in 3rd instar larvae under control conditions, with Pavarotti or Trc RNAi driven by elavGal4, or both RNAis together. Scale bar = 50 µm. n = 15–30 cells from at least 4 animals. Ctrl = 2549 ± 108 µm, Upper 95% CI = 2771, Lower 95% CI = 2328, Pav RNAi = 3803 ± 141 µm, 4099, 3507,

*Figure 5 continued on next page*

*Figure 5 continued*

Trc RNAi = 3577 ± 155 µm, 3904, 3250, Pav and Trc RNAi = 3455 ± 257 µm, 4005, 2905. (D) Pavarotti and Trc depletion cause an increase in dendritic length compared to control. Depletion of both proteins together has a non-additive effect compared with either RNAi. Ctrl vs Pav RNAi p<0.0001, Ctrl vs Trc RNAi, p<0.0001, Ctrl vs double RNAi p=0.0006, PavRNAi vs Trc RNAi p=0.76, Pav RNAi vs double RNAi p=0.48, Trc RNAi vs Double RNAi p=0.96. One-way ANOVA with Tukey's post hoc correction.

The online version of this article includes the following source data and figure supplement(s) for figure 5:

**Source data 1.** Sliding rates in neurons upon Trc, Pavarotti, or Trc and Pavarotti knockdown.
**Source data 2.** Dendritic length of DA neurons.
**Figure supplement 1.** Analysis of branch points per cell in DA neurons.
**Figure supplement 1—source data 1.** Number of branch points of DA neurons.

biochemically that *Drosophila* Pavarotti is phosphorylated at this site by Trc in neurons. This extends previous reports to show this pathway is conserved between humans and *Drosophila* (*Fesquet et al., 2015*). Our in vitro data suggest that these two proteins act together to regulate sliding in developing cultured neurons. In vivo, we demonstrate that these two proteins work in concert to regulate neuronal development. Depletion of either Trc or Pavarotti leads to increased dendrite length in class IV DA neurons. Depletion of both these proteins simultaneously has no additive effect either with regards to sliding or dendrite outgrowth, therefore these proteins act in a common pathway. Notably, this is the first report of Pavarotti regulating dendrite development in *Drosophila*. Previous work has shown Pavarotti prevents axon overgrowth (*Del Castillo et al., 2015*) and reports in mammalian systems have suggested roles in both compartments (*Lin et al., 2012*).

Whilst protein translation presents a clear alternative in regulating protein activity, we favor a phosphorylation model. Pavarotti expression is inhibited by Toll-6-FoxO signaling and Toll-6-FoxO mutants have increased microtubule stability (*McLaughlin et al., 2016*). However, phosphorylation would provide more dynamic method for modulating Pavarotti. Moreover, phosphorylation would provide tighter spatial regulation which could be necessary in inhibiting sliding in primary neurites while secondary processes are still developing. Indeed, examining the subcellular distribution of Pavarotti showed clear differences in microtubule association based on Trc-mediated phosphorylation state. We found Pavarotti localized more robustly to microtubules in the presence of Trc and that the phospho null mutant had a decreased ability to associate with microtubules. Upon association with microtubules, Pavarotti acts as a crosslinker and inhibits microtubule sliding. These observations are consistent with our sliding data – preventing Pavarotti phosphorylation by Trc upregulated microtubule sliding. It is possible that Pavarotti phosphorylation inhibits kinesin-1-mediated microtubule sliding initially, and is subsequently regulated at the protein translational level.

## Pavarotti requires 14-3-3 proteins to Brake microtubule sliding in S2 cells

Extending our hypothesis that mitotic mechanisms regulating Pavarotti may be prevalent in neurons, we chose to investigate the role of 14-3-3s in microtubule sliding. 14-3-3s are conserved acidic proteins which bind phospho-threonine and phospho-serine residues (*Cornell and Toyo-oka, 2017*). Interaction and complex formation with phosphorylated proteins to facilitate cytoskeleton remodelling and axon extension has been described multiple times (*Cornell and Toyo-oka, 2017*; *Taya et al., 2007*). In *C. elegans* in mitosis, 14-3-3s have been shown to bind to the centralspindlin complex when Zen-4 (the *C. elegans* orthologue of Pavarotti/MKLP1) is phosphorylated at S710 (equivalent of Pavarotti S745) (*Douglas et al., 2010*). Here, we show by co-immunoprecipitation experiments that Trc-mediated phosphorylation at this site promotes

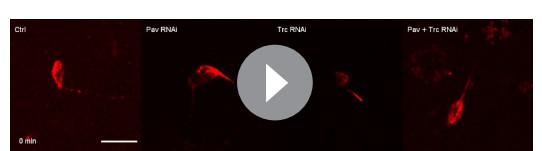

**Video 9.** Pavarotti and Trc control microtubule sliding together in neurons. Timelapse imaging of photoconverted microtubules in *Drosophila* cultured neurons under control conditions or with depletion Fry. Microtubules were labeled with UAS tdMaple alpha Tubulin 84b under control of a motor-neuron-specific D42 gal4 driver. Trc was depleted with UAS Fry RNAi under control of a motor-neuron-specific D42 gal4 driver. 2 frames per minute. Scale bar 10 µm.
https://elifesciences.org/articles/52009#video9

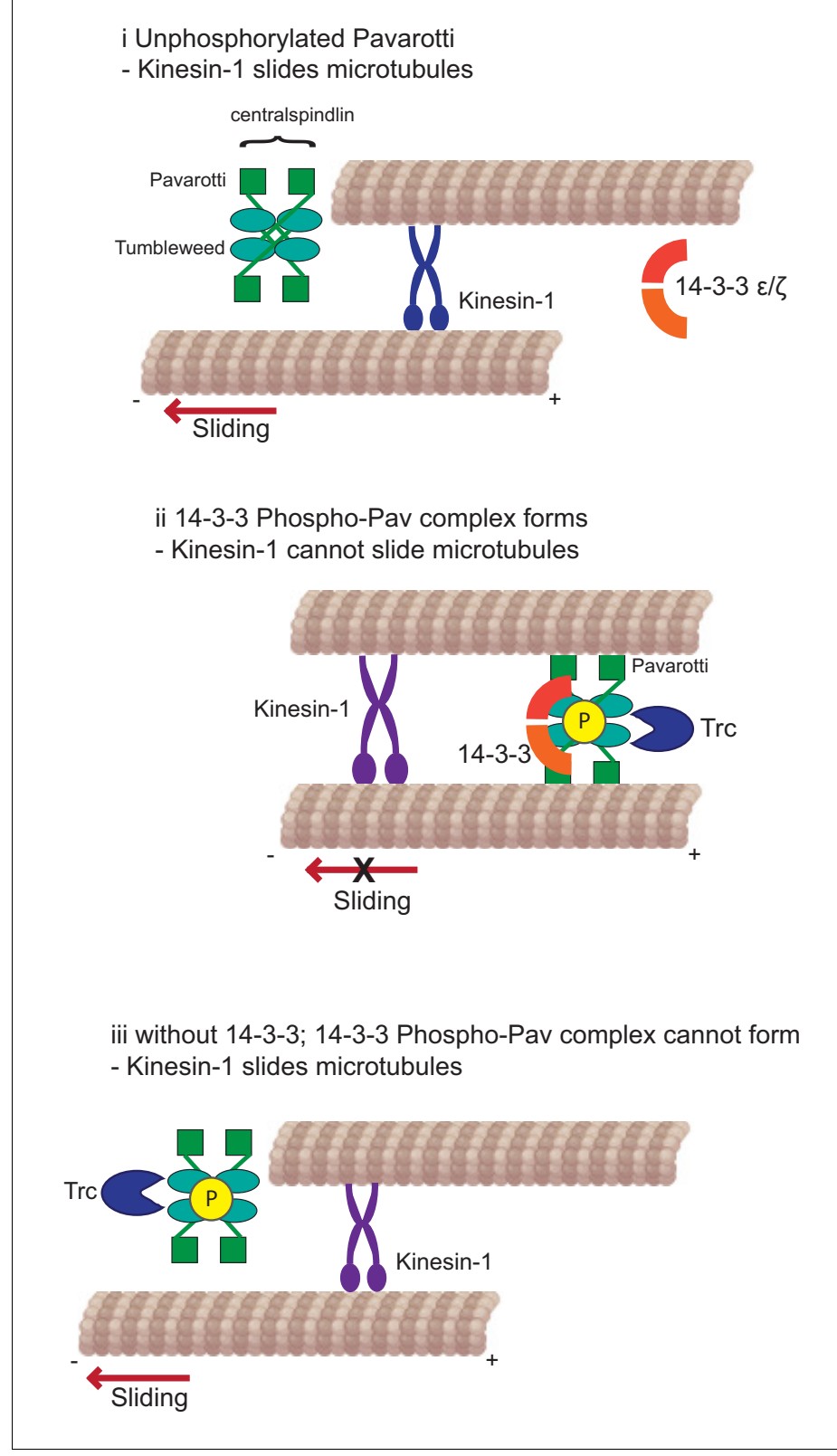

**Figure 6.** Kinesin-1 slides microtubules along one another to facilitate neurite outgrowth. The kinase Trc inhibits this process by phosphorylation of Pavarotti – a part of the centralspindlin complex with tumbleweed. Phosphorylated Pavarotti forms a complex with 14-3-3 proteins and associates with microtubules. Under these

*Figure 6 continued on next page*

*Figure 6 continued*

conditions, microtubules are cross-linked and can no longer undergo sliding by kinesin-1. Therefore, neurite outgrowth is inhibited.

formation of a complex between Pavarotti and 14-3-3s, in good agreement with previous data (*Fesquet et al., 2015*). This complex has previously been proposed to prevent stable microtubule binding in vitro. Interestingly, our data in S2 cells are not in full agreement with in vitro data. Previous studies have proposed phosphorylation by aurora B kinase at S708/S743 in *Drosophila* as a way to release 14-3-3 binding from centralspindlin. Perhaps, this phosphorylation is differently regulated during mitosis, where the centralspindlin complex must accumulate at microtubule tips, and interphase, where the microtubule cytoskeleton must be stabilized (*Douglas et al., 2010*). Our data suggest 14-3-3 interaction with Pavarotti promotes stable microtubule association. Considering our data together with previously published work, perhaps this is a facet of microtubule sliding regulation that is differentially controlled between mitosis and interphase. Beyond interphase, the role of 14-3-3 proteins in sliding inhibition and neurite outgrowth warrants further investigation in neurons.

Here, we present a microtubule-based mechanism for Trc and Pavarotti controlling neurite development and show how the kinase Trc directs Pavarotti intracellular localization. Beyond this, it would be of interest to investigate a concurrent role for actin remodelling in process outgrowth. Pavarotti is a component of the centralspindlin complex (*Mishima et al., 2002*). The other component, tumbleweed/MgcRacGAP, gives an axon overextension phenotype upon depletion, like Pavarotti (*Goldstein et al., 2005*) and the expression levels of each are dependent on the other (*Del Castillo et al., 2015*). However, there are no described Trc kinase phosphorylation sites in tumbleweed. Therefore, while our experiments have focussed on Pavarotti phosphorylation, it is likely the downstream effects described here are a result of the centralspindlin complex. MgcRacGAP is a major orchestrator of RhoA signaling via Pebble/RhoGEF (*Basant and Glotzer, 2018*). This RhoGEF promotes formation of the cytokinetic furrow via actin assembly. Further, Trc has been proposed to inhibit Rac activity in a kinase-dependent manner (*Emoto et al., 2004*). It has recently been demonstrated that the centralspindlin complex acts as a signaling hub to spatially and temporally regulate contraction of the actin cortex (*Verma and Maresca, 2019*). Might these signaling pathways be acting in developing neurites to tailor their development in addition to mechanical regulation via microtubule sliding?

The data presented here raise new questions beyond regulating microtubule sliding regulation. These data along with previous work from our group and others show a role for Pavarotti in controlling both axon and dendrite outgrowth (*Del Castillo et al., 2015*; *Lin et al., 2012*). This is consistent with our findings that kinesin-1 mediated microtubule sliding is necessary for proper development of both these compartments (*Winding et al., 2016*). Further, does microtubule sliding play a role in specifying axon formation? A crucial and distinctive feature of axons and dendrites is that of their microtubule polarity – axonal microtubules have a uniform, plus end out microtubule orientation, whereas dendritic microtubules are of mixed polarity or uniformly minus ends out (*Baas et al., 1988*; *Stone et al., 2008*). Kinesin-6 has been previously proposed to confer dendritic identity via transport of minus ends distal microtubules into dendrites and away from axons (*Lin et al., 2012*; *Yu et al., 2000*). Does the regulation of microtubule sliding via Pavarotti/kinesin-6 phosphorylation contribute to the microtubule polarity of nascent processes? How might these regulatory processes change over the course of axonal and dendritic development?

As well as neurite initiation, microtubule sliding occurs during axon regeneration (*Lu et al., 2015*). After axon or dendrite severing, in vitro or in vivo, large-scale rearrangements of the microtubule cytoskeleton are observed (*Lu et al., 2015*; *Stone et al., 2010*). This is in contrast to in mature neurons where sliding is silenced. In this case, it could be of great interest to exploit our suggested mechanism of Pavarotti phosphorylation. Notably, the kinase Trc may be a promising candidate for chemical inhibition. Would chemical inhibition or silencing of the kinase Trc deplete the pool of phospho-Pav and prolong the time period during which microtubule sliding was upregulated? Could this, in turn, facilitate neurite regeneration after injury? Further work will be required to address any potential for modulating Trc activity in neuronal regeneration.

# Materials and methods

## Key resources table

| Reagent type (species) or resource | Designation | Source or reference | Identifiers | Additional information |
|---|---|---|---|---|
| Genetic reagent (*D. melanogaster*) | *w; elav-Gal4 (III)* | C. Doe University of Oregon | | |
| Genetic reagent (*D. melanogaster*) | *yw; wg^(Sp)/CyO; Dr^(Mio)/TM3, Sb* | E. Ferguson, University of Chicago | | |
| Genetic reagent (*D. melanogaster*) | *yw; ppk-CD4-tdtomato (II)* | Bloomington *Drosophila* Stock Center (BDSC) | Stock number 35844; FBst0035844; RRID:BDSC_35844 | Fly base genotype: w[1118]; P{ppk-CD4-tdTom}4a |
| Genetic reagent (*D. melanogaster*) | *w; D42-Gal4 (III)* | BDSC | Stock number 8816; FBst0008816; RRID:BDSC_8816 | Fly base genotype: w*; P{GawB}D42 |
| Genetic reagent (*D. melanogaster*) | *y sc v; UAS-Trc-RNAi* | BDSC | Stock number 41591; FBst0041591; RRID:BDSC_41591 | Fly base genotype y[1] sc*v[1] sev[21]; P{TRiP.GL01127}attP2 |
| Genetic reagent (*D. melanogaster*) | *y sc v; UAS-Pav-RNAi* | BDSC | Stock number 42573; FBst0042573; RRID:BDSC_42573 | Fly base genotype: y[1] v[1]; P{TRiP.HMJ02232}attP40 |
| Genetic reagent (*D. melanogaster*) | *y sc v; UAS-Fry-RNAi* | BDSC | Stock number 60103; FBst0060103; RRID:BDSC_60103 | Fly Base Genotype: y[1] sc*v[1] sev[21]; P{TRiP.HMC05097}attP40 |
| Genetic reagent (*D. melanogaster*) | *w; UASp-tdMaple3-alpha tubulin 84b* | This lab | | Described in *Lu et al. (2016)* Generated by injection of pUASp tdMaple3-alpha tubulin 84b. A second chromosome insertion was used in this study. |
| Genetic reagent (*D. melanogaster*) | *yw; UASp-tdMaple3-alpha tubulin 84b; UAS-Trc-RNAi* | This lab | | Generated using an insertion on the second chromosome |
| Transfected construct (*D. melanogaster*) | pMT EOS tubulin | This lab | | Described in *Barlan et al. (2013)*. For S2 cell expression of EOS tubulin. |
| Transfected construct (*D. melanogaster*) | pMT-BFP Trc WT | This lab | | S2 cell expression of Trc. Generated from pUASt-Trc WT using EcoRI and NotI, a kind gift from P. Adler. |
| Transfected construct (*D. melanogaster*) | pMT-BFP Trc CA (T253E) | This lab | | S2 cell expression of Trc. Generated from pUASt-Trc T253E (a kind gift from P. Adler) and pMT vector using EcoRI and NotI. |
| Transfected construct (*D. melanogaster*) | pMT-BFP Trc kinase dead (K122A) | This lab | | S2 cell expression of Trc. Generated from pUASt-Trc K122A (a kind gift from P. Adler) and pMT vector using EcoRI and NotI. |
| Transfected construct (*D. melanogaster*) | pMT-GFP Pavarotti | This lab | | S2 cell expression. Generated from pMT-BFP Pavarotti (*Del Castillo et al., 2015*) |
| Transfected construct (*D. melanogaster*) | pMT-GFP Pavarotti S745A | This lab | | S2 cell expression. Generated by site directed mutagenesis of WT. |

*Continued on next page*

Continued

| Reagent type (species) or resource | Designation | Source or reference | Identifiers | Additional information |
|---|---|---|---|---|
| transfected construct (*D. melanogaster*) | pEGFP Pavarotti WT | This lab | | Mammalian expression. Pavarotti constructs were subcloned into pEGFP-C1 using EcoRI and SalI. |
| Transfected construct (*D. melanogaster*) | pEGFP Pavarotti S745A | This lab | | Mammalian expression. Pavarotti constructs were subcloned into pEGFP-C1 using EcoRI and SalI. |
| transfected construct (*D. melanogaster*) | pcDNA Trc T253E | This lab | | Mammalian expression. Trc T253E was subcloned into pcDNA 3.1+ using HindIII and NotI. |
| Transfected construct (*D. melanogaster*) | pMT mCherry tubulin | This lab | | For S2 cell expression. Described in **del Castillo et al. (2015)** |
| Cell line (*D. melanogaster*) | S2 cells | DGRC | FlyBase Report: FBtc0000006 | Cell line maintained in this lab. |
| Cell line (*Homo-sapiens*) | HEK 293 FT | ATCC | RRID:CVCL_0045; ATCC: CRL-1573 | Cell line maintained in this lab. |
| Antibody | Single chain anti-GFP, GFP-Trap-M | Chromotek | | |
| Antibody | Anti Pavarotti (Rabbit polyclonal) | Scholey lab | | Western blot 1:1000 |
| Antibody | Anti Trc (Rabbit polyclonal) | Emoto lab | | Western blot 1:1000 |
| Antibody | Anti Tubulin (Rabbit polyclonal) | This lab | | Affinity purified from immunized rabbit serum. Western blot 1:1000 |
| Antibody | Anti phospho-Pavarotti S710 (Rabbit polyclonal) | Mishima lab | | Western blot 1:1000 |
| Antibody | Anti GFP (Rabbit polyclonal) | This lab | | Affinity purified from immunized rabbit serum. Western blot 1:1000 |
| Antibody | Anti Furry (Rabbit polyclonal) | Adler lab | | Immunofluorescence 1:250 |
| Antibody | Anti Hsc 70 (Goat polyclonal) | Santa cruz | K-19, RRID: AB_2120291 | Western blot 1:100 |
| Antibody | Anti 14-3-3ζ (Rabbit polyclonal) | Proteintech group | cat no. 14503–1-AP; RRID:AB_2218096 | Western blot 1:250 |
| Antibody | HRP conjugated Anti Rabbit | Jackson | | Western blot 1:10,000 |
| Antibody | HRP conjugated Anti Goat | Jackson | | Western blot 1:10,000 |

## Fly stocks

Flies were maintained at room temperature (24 ~ 25 C) on regular cornmeal food (Nutri-Fly, Bloomington Formulation), supplemented with dry active yeast. dsRNA generation dsRNAs were generated using the sequences described in the supplementary table with the T7 sequence TAATACGAC TCACTATAGGG at the 5' end.

## Cell culture

*Drosophila* S2 cells were maintained in Insect-Xpress medium (Lonza) at 25˚C. These cells came directly from DGRC and their identity was not subsequently confirmed. Transfections were carried out with Effectene (Qiagen) according to the manufacturer's instructions. dsRNA was added to cells on days 1 and 3 and imaging was carried out on day 5. HEK 293 FT cells were maintained in DMEM (Sigma Aldrich) supplemented with Penicillin Streptomycin and 10% FBS at 37˚C and 5% $CO_2$. HEK cells were transfected by Calcium Phosphate precipitation with 5 µg DNA. Cells were confirmed mycoplasma negative by Lonza mycoalert mycoplasma detection kit.

Primary neuronal cultures were prepared by dissection of brains from 3rd instar larvae and dissociation of tissue using liberase (Roche). Cells were plated on ConA coated glass coverslips and maintained in Schneiders medium supplemented with 20% FBS, 5 µg/ml insulin, 100 µg/ml Pen-Strep, 50 µg/ml Gentamycin and 10 µg/ml Tetracycline. For sliding assays, larvae were cultured at 29˚C and imaged 1 hr after plating.

## Immunoprecipitation and western blotting

Co-Immunoprecipitation from HEK 293 cells was carried out in coIP buffer (50 mM Tris pH 7.5, 150 mM NaCl, 1.5% Triton X-100, 1 mM EDTA, 1 mM PMSF, 20 µg/ml Chymostatin, Leupeptin, Pepstatin, 1 mM $NaVO_3$). Cells were lysed, debris was pelleted by centrifugation, and the soluble fraction was incubated with single chain anti GFP antibody (GFP-binder) (GFP-Trap-M; Chromotek) conjugated to sepharose beads. Samples were washed 3x in lysis buffer and boiled in 5x laemmli buffer prior to loading on 10% acrylamide gels for electrophoresis. For phosphorylation experiments, GFP Pavarotti was enriched by GFP pull down in RIPA buffer (50 mM Tris pH 7.4, 150 mM NaCl, 1% Triton, 0.5% Na-Deoxycholate, 0.1% SDS, 1.5 mM $NaVO_3$, 1 mM PMSF, 20 µg/ml Chymostatin, Leupeptin, Pepstatin, 1 mM $NaVO_3$) and samples were processed as for co-immunoprecipitation For phosphorylation experiments from brain lysate, around 50 3rd instar larvae brains were dissected per condition and lysed in RIPA buffer. The soluble fraction was subjected to IP with anti-Pavarotti antibody or rabbit IgG. Protein-G Dynabeads (Invitrogen) were blocked in 1% BSA in TBS-T and added to lysates. IPs were washed 3x in RIPA buffer and boiled in 5x laemmli buffer prior to loading on 8% acrylamide gels. To assess efficient knockdown of proteins, S2 cells were lysed directly in sample buffer and boiled. After electrophoresis, transfer onto nitrocellulose membrane was carried out and blocking was performed in 4% milk in PBS-T. For phospho- specific antibody, blocking was carried out with 3% BSA in TBS-T. Western blotting was performed using advansta western bright quantum substrate and Licor Imagequant system. For quantification of western blots, Licor Image Studio software was used. In each case, phosphorylated Pavarotti levels were compared to total Pavarotti levels. Then this ratio was normalized to that of the control condition. In this way, any difference in immunoprecipitated Pavarotti levels is accounted for between the two conditions.

## Fixed imaging

For subcellular localization analysis of Pavarotti, S2 cells were plated on ConA coated coverslips and allowed to attach. Cells were then extracted in 30% glycerol, 1%triton, 1 uM taxol in BRB80 for 3 min and imaged directly.

## Microscopy and photoconversion

To image dissociated neuronal cultures by phase contrast we used an inverted microscope (Eclipse U2000; Nikon Instruments) equipped with 60x/1.40 N.A objective and a CoolSnap ES CCD camera (Roper Scientific) and driven by Nikon Elements software.

To image *Drosophila* S2 cells and primary neurons, a Nikon Eclipse U200 or Ti2 inverted microscope with a Yokogawa CSU10 or Yokogawa W1 spinning disk confocal head, Nikon Perfect Focus system, and 100×/1.45 N.A. objective was used. Images were acquired with an Evolve EMCCD or Prime 95B (Photometrics) using Nikon NIS-Elements software (AR 4.00.07 64-bit). S2 cells expressing tdEOS-tagged Tubulin were plated in Xpress with 2.5 µM cytoD and 40 nM taxol. For photoconversion of tdEOS-tagged Tubulin in sliding assays, we applied 405 nm light from a light-emitting diode light source (89 North Heliophor) for 5 s. The 405 nm light was constrained to a small circle with an adjustable diaphragm, therefore only a region of interest within the cell was photoconverted. After photoconversion, images were collected every minute for > 10 min.

To image *Drosophila* DA neurons, larvae were immobilized between a slide and a coverslip. A Nikon Eclipse U200 inverted microscope with a Yokogawa CSU10 spinning disk confocal head, 20×/ 0.75 N.A. objective was used. Images were acquired with an Evolve EMCCD (Photometrics) using Nikon NIS-Elements software (AR 4.00.07 64-bit). Images were analyzed in FIJI using the 'tubeness' and 'skeletonize' plugins. Branch point analysis was carried out in Nikon NIS-Elements software (AR 5.200 64-bit) using the detect branches function in the general analysis three module.

## Microtubule sliding analysis

Analysis was carried out as previously described. Briefly, time-lapse movies of photoconverted microtubules were bleach-corrected and thresholded and the initial photoconverted zone was identified. The number of pixels corresponding to MTs was measured in total or outside the initial zone for each frame. The motile fraction (defined as $MTs^{outside\_initial\_zone}/MTs^{total}$) was plotted against time and the slope of the linear portion was calculated to represent microtubule sliding rate. For analysis of microtubule sliding in neurons, images were bleach corrected and denoised using despeckle in FIJI. Movies were then processed using the WEKA trainable segmenter in Fiji to generate probability maps of photoconverted microtubules (*Arganda-Carreras et al., 2017*). Probability maps were thresholded and analyzed in the same way as S2 cells.

## Statistical analysis and data presentation

Data are presented as mean ± standard error. Statistical analysis was carried out in GraphPad. Data were analyzed using student's T-test or one-way ANOVA with Sidak's post-hoc correction for multiple comparisons. Data are collected from at least three replicates. Statistical significance is presented as *p<0.05, **p<0.01, ***p<0.001. Data in Figure legends are presented as mean ± standard error, Upper 95% Confidence Interval, Lower 95% Confidence interval.

# Acknowledgements

This work was supported by NIH grants R01 GM052111 and R35 GM131752 to V Gelfand. We thank members of the Gelfand lab and M Glotzer for helpful discussion. We thank the Bloomington Stock Center (supported by NIH P40OD018537) for fly stocks. We thank M Mishima for phospho-specific Pavarotti antibodies, K Emoto for Trc antibody and we thank P Adler for Trc DNA constructs. We thank D Kirchenbuechler of the Northwestern University Center for Advanced Microscopy for help with analysis in *Figure 5—figure supplement 1B*.

# Additional information

## Funding

| Funder | Grant reference number | Author |
| --- | --- | --- |
| National Institutes of Health | R01 GM052111 | Vladimir I Gelfand |
| National Institutes of Health | R35 GM131752 | Vladimir I Gelfand |

The funders had no role in study design, data collection and interpretation, or the decision to submit the work for publication.

## Author contributions

Rosalind Norkett, Conceptualization, Formal analysis, Investigation, Methodology; Urko del Castillo, Resources, Data analysis data interpretation, manuscript editing; Wen Lu, Resources, Generation of flies used in the study, manuscript editing; Vladimir I Gelfand, Conceptualization, Supervision, Funding acquisition

## Author ORCIDs

Vladimir I Gelfand (iD) https://orcid.org/0000-0002-6361-2798

Decision letter and Author response
Decision letter https://doi.org/10.7554/eLife.52009.sa1
Author response https://doi.org/10.7554/eLife.52009.sa2

## Additional files

### Supplementary files
- Source code 1. IJM macro used for sliding analysis.
- Supplementary file 1. Primer sequences used for dsRNA generation.
- Transparent reporting form

### Data availability
All data generated or analyzed during this study are included in the manuscript and supporting files.

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
