## [Decision Letter]

**Acceptance summary:**

Neuronal process formation must be precisely regulated such that neurite outgrowth is robust in developing neurons in order to form proper communication networks and limited in mature neurons once neurites have reached their appropriate targets. The study here uses a combination of cell-based systems including *Drosophila* S2 cells, primary neuronal cultures, and sensory neurons to investigate microtubule-based mechanisms for neurite outgrowth. In doing so, they uncover a new mechanism for how neuronal processes are limited to ensure proper development of the nervous system.

**Decision letter after peer review:**

Thank you for submitting your article "Ser/Thr kinase Trc controls neurite outgrowth by modulating microtubule-microtubule sliding" for consideration by *eLife*. Your article has been reviewed by three peer reviewers, and the evaluation has been overseen by a Reviewing Editor and Suzanne Pfeffer as the Senior Editor. The reviewers have opted to remain anonymous. The reviewers have discussed the reviews with one another and the Reviewing Editor has drafted this decision to help you prepare a revised submission.

Summary:

Previous work from the Gelfand group has shown that microtubule sliding, driven by kinesin-1, promotes neurite formation and extension, and the kinesin-6 motor Pavarotti inhibits these processes in *Drosophila* neurons. In the present study, the authors describe a regulatory pathway that leads to inhibition of kinesin-1-dependent microtubule sliding, thereby limiting the extension of neuronal processes. The principle finding is that the Ser/Thr kinase Trc blocks microtubule sliding by promoting the activity of Pavarotti. The authors demonstrate that Trc phosphorylates Pavarotti on S745, and this in turn promotes a biochemical interaction between Pavarotti and 14-3-3 proteins. They conclude that this complex formation leads to microtubule binding and inhibition of microtubule sliding. The authors make use of a microtubule sliding assay in cultured cells, biochemistry, and an in vivo dendrite morphology assay to support their findings. The evidence that Trc regulates microtubule sliding and neurite outgrowth in cultured neurons is strong, and the authors show that loss of either Pav or Trc results in an increase in dendrite length in Class IV sensory neurons. However, much of the study, including the experiments arguing that the phosphorylation state of Pav is important, that Trc is the relevant kinase, and that 14-3-3 is important for this pathway, is done in cultured S2 cells. It is not clear if microtubule dynamics in these cells are regulated comparably to those in developing neurons, thus it will be important for the authors to demonstrate that the results obtained in S2 cells regarding Trc phosphorylation of Pav and the role of 14-3-3 in the regulatory pathway are recapitulated in an appropriate neuronal cell type.

Essential revisions:

1) The data that Trc phosphorylates Pav and promotes its ability to limit sliding in S2 cells are convincing. However, there is little evidence that this phosphorylation is relevant in neurons. To address the proposed role of Trc as a neuronal regulator of Pav-mediated microtubule braking, the authors should carry out the same experiments presented in Figures 2 and 3 in neurons to show that Pav is phosphorylated on S745, that the kinase activity of Trc is important for its function, and that the Pav S745A mutant does not block sliding. In del Castillo et al., 2015, the authors addressed Pav function in third instar motor axons, which may be a good option for addressing the role of Trc in Pav regulation in a relevant cell type.

2) Contrary to the authors' claim that 14-3-3 is a key regulator of Pav's ability to regulate sliding, 14-3-3 knockdown in S2 cells has no sliding phenotype on its own. The authors only show that knockdown of 14-3-3 impedes the ability of Pav overexpression to inhibit sliding. In light of this, they may consider revising their model of 14-3-3 function.

3) Related to the above point, is Pavarotti phosphorylated and bound to 14-3-3 in neurons? In the paper, the authors demonstrate that Pavarotti is phosphorylated on S745 in HEK293T cells (and more so when Trc is co-expressed). Similarly, evidence to support a functional interaction between Pavarotti and 14-3-3 comes from experiments performed in non-neuronal cells. The authors should demonstrate that Pavarotti is phosphorylated and interacts with 14-3-3 in neurons.

4) The authors propose in their model that 14-3-3 binding to the phospho-epitope promotes the association of Pavarotti with microtubules and its crosslinking and braking activity. However, Douglas et al., 2010, published that 14-3-3 binding to the S710 residue in human kinesin-6 reduced the MT binding and bundling activity of centralspindlin. The authors should account for this discrepancy in the text.

5) Figure 5: The authors show that Pav or Trc knockdown leads to an increase in the total length of dendrites of Class IV neurons. It has been previously reported that Trc knockdown increases branching (Emoto et al., 2004). It is unclear why increased MT sliding would cause an increase in branching as opposed to dendrite growth. Do the authors observe an increase in branching after Pav knockdown? In the images presented, the effect of Pav depletion looks qualitatively different than the effect of Trc depletion; Pavarotti depletion does appear by eye to produce more secondary and tertiary branches than Trc depletion. Is this the case? These parameters should be quantified and discussed.

6) Figure 5: If possible, these analyses should be carried out at an earlier time point to demonstrate that this phenotype is the result of a defect in initial dendrite outgrowth.

7) In Figure 1G (and Video 2), there does not appear to be a clear difference in sliding between the two conditions from the images/video. This should be addressed.

8) There are a number of instances (listed below) throughout the manuscript where knockdown efficiencies and expression levels should be assessed: a) Furry knockdown in Figure 2; b) Pav depletion and expression experiments in Figure 3C, D; c) PavGFP over-expression in Figure 4; d) Knockdown efficiencies of Trc and Pav in the DA neurons in Figure 5. This is a beautiful cell type and assay but it needs to be accompanied by some assessment (IF, WB, RT-qPCR) that the RNAi is working in these cells.

9) The measurements of the% microtubule area shown in Figures 3 and 4 are unclear, especially as presented as a single merged image. Please show each individual channel and highlight the area that you designate as "Pav positive". Pav is nuclear in interphase and, while the images are difficult to assess, it appears that the nuclear signal is evident in some of them. How is this taken into account when evaluating the area that is measured?

10) The authors reference a constitutively-active Trc in the subsection “Trc kinase activity is necessary to control microtubule sliding”, and refer in to data showing that this construct reduces sliding comparably to WT Trc; however, these data are not presented in Figure 2.

11) The co-IPs in Figure 4A should be quantified. In addition, the experiment shown in 4A is not currently described in sufficient detail so that readers can understand exactly what was done. Please provide a more detailed explanation.

12) Very little is mentioned in the manuscript about the other component of centralspindlin – MgcRacGAP. The authors should clarify if they believe this mechanism is being mediated by the centralspindlin complex. Part of the source of the confusion also stems from the model figure that appears to show a tetramer of Pavarotti – is that the intention or is this supposed to be a Pavarotti dimer and a Tumbleweed dimer? If it is centralspindlin then the complex should be labeled appropriately and the schematic should be changed accordingly (at least different colors for the constituents).

[Editors' note: further revisions were suggested prior to acceptance, as described below.]

Thank you for resubmitting your work entitled "Ser/Thr kinase Trc controls neurite outgrowth by modulating microtubule-microtubule sliding" for further consideration by *eLife*. Your revised article has been evaluated by Suzanne Pfeffer as the Senior Editor, and a Reviewing Editor.

The manuscript has been significantly improved, but there are a few remaining issues that need to be addressed before acceptance, as outlined below:

1) Much of the new data relates to extending the findings from *Drosophila* S2 cells further into neurons. To do so the authors have added additional sliding data from cultured neurons to Figure 2 (and Figure 2—figure supplement 1) and Figure 5 that agree with the results from S2 cells. They also show that there is a reduction of Pavarotti phosphorylation on S745 following depletion of Trc in fly brains in Figure 3. While the levels of pS745-Pav are quantified, the total Pav levels are not. The authors need to quantify the bands in the lower blot of Figure 3J and report the ratio of pS745 to total Pavarotti.

2) The author's claim that 14-3-3 is required for regulation of microtubule sliding is still not explicitly supported by the data presented. Figure 4B and 4C clearly show that there is no effect on sliding when 14-3-3 is inhibited. In light of this, the language in the Abstract should be modified.

3) "In vitro we show that these two proteins act together to regulate sliding in developing cultured neurons." This sentence needs to be re-written to more clearly indicate that the in vitro studies support a model in which the two proteins act together to regulate sliding in developing cultured neurons.

---

## [Author Response]

Summary:Previous work from the Gelfand group has shown that microtubule sliding, driven by kinesin-1, promotes neurite formation and extension, and the kinesin-6 motor Pavarotti inhibits these processes in *Drosophila* neurons. In the present study, the authors describe a regulatory pathway that leads to inhibition of kinesin-1-dependent microtubule sliding, thereby limiting the extension of neuronal processes. The principle finding is that the Ser/Thr kinase Trc blocks microtubule sliding by promoting the activity of Pavarotti. The authors demonstrate that Trc phosphorylates Pavarotti on S745, and this in turn promotes a biochemical interaction between Pavarotti and 14-3-3 proteins. They conclude that this complex formation leads to microtubule binding and inhibition of microtubule sliding. The authors make use of a microtubule sliding assay in cultured cells, biochemistry, and an in vivo dendrite morphology assay to support their findings. The evidence that Trc regulates microtubule sliding and neurite outgrowth in cultured neurons is strong, and the authors show that loss of either Pav or Trc results in an increase in dendrite length in Class IV sensory neurons. However, much of the study, including the experiments arguing that the phosphorylation state of Pav is important, that Trc is the relevant kinase, and that 14-3-3 is important for this pathway, is done in cultured S2 cells. It is not clear if microtubule dynamics in these cells are regulated comparably to those in developing neurons, thus it will be important for the authors to demonstrate that the results obtained in S2 cells regarding Trc phosphorylation of Pav and the role of 14-3-3 in the regulatory pathway are recapitulated in an appropriate neuronal cell type.

We thank the reviewers for their careful consideration of our work. We agree with the reviewers’ overall assessment that our work relied heavily on data from cultured S2 cells. Therefore, as suggested by the reviewers, we have expanded our data to include the role of Trc in neurons. We show that it is not just the Trc kinase expression level that regulates sliding, but more specifically, Trc kinase activity is required to regulate microtubule sliding (Figure 2E, F). We also show that, related to the development of Class IV sensory neuron dendrites (Figure 5C, D), microtubule sliding is regulated by Pavarotti and Trc together (Figure 5A, B). These data support our hypothesis that microtubule sliding is in S2 cells and developing neurons are regulated by similar mechanisms. Significantly, we now include biochemistry from *Drosophila* larvae brain to demonstrate Trc dependent Pavarotti phosphorylation (Figure 3J). In addition, we greatly expand on our demonstration of relevant knockdown throughout the manuscript. For an itemized description of our revisions, please see below.

Essential revisions:1) The data that Trc phosphorylates Pav and promotes its ability to limit sliding in S2 cells are convincing. However, there is little evidence that this phosphorylation is relevant in neurons. To address the proposed role of Trc as a neuronal regulator of Pav-mediated microtubule braking, the authors should carry out the same experiments presented in Figures 2 and 3 in neurons to show that Pav is phosphorylated on S745, that the kinase activity of Trc is important for its function, and that the Pav S745A mutant does not block sliding. In del Castillo et al., 2015, the authors addressed Pav function in third instar motor axons, which may be a good option for addressing the role of Trc in Pav regulation in a relevant cell type.

We agree with the reviewers’ assessments. We have now expanded our experiments in neurons beyond demonstrating that Trc protein level regulates sliding in these cells. To address the requirement for Trc kinase activity to inhibit microtubule sliding (as in Figure 2), we have taken advantage of the fact that Trc kinase activity requires the presence of another protein – Furry (Fry). We have carried out sliding experiments in cultured neurons upon knockdown of Furry/Fry, as well as S2 cells. The results of these experiments are now shown in Figure 2E and F. We demonstrated that Fry knockdown in both S2 cells and in neurons increases microtubule sliding. The control experiments demonstrate that knock-down of Fry does not change Trc protein level. Taken together, these data demonstrate that the kinase activity of Trc is required to regulate sliding in both S2 cells and in neurons.

We have now also demonstrated that Trc and Pavarotti act together to regulate microtubule sliding in neurons, extending our experiments in Figure 3A. Unfortunately, the experiment that the reviewer suggested in 3^rd^ instar motor axons did not give us an obvious phenotype, most likely because Trc is a kinase, so a much higher level of knock-down is required for observing an effect. Instead, we used a different approach. We show in the updated Figure 5A that knockdown of both of these proteins in neurons increases microtubule sliding to similar degree to each RNAi individually. Together with in vivoneurite outgrowth data in Figure 5C and D, these data support the conclusion that Trc and Pavarotti tailor neurite outgrowth together, by controlling microtubule sliding.

2) Contrary to the authors' claim that 14-3-3 is a key regulator of Pav's ability to regulate sliding, 14-3-3 knockdown in S2 cells has no sliding phenotype on its own. The authors only show that knockdown of 14-3-3 impedes the ability of Pav overexpression to inhibit sliding. In light of this, they may consider revising their model of 14-3-3 function.

We fully agree with the reviewers’ suggestion. Therefore, we have amended our model to reflect the fact 14-3-3 RNAi only has a phenotype upon Pavarotti overexpression. Please see the revised summary figure in the Discussion section. We also expand our Discussion with regards to these data and now postulate that, while 14-3-3 appears to be a regulator of microtubule sliding, there appears to be some divergence between mitotic mechanisms and interphase mechanisms. Please see the first paragraph of the subsection “Pavarotti requires 14-3-3 proteins to brake microtubule sliding in S2 cells”, in the Discussion.

3) Related to the above point, is Pavarotti phosphorylated and bound to 14-3-3 in neurons? In the paper, the authors demonstrate that Pavarotti is phosphorylated on S745 in HEK293T cells (and more so when Trc is co-expressed). Similarly, evidence to support a functional interaction between Pavarotti and 14-3-3 comes from experiments performed in non-neuronal cells. The authors should demonstrate that Pavarotti is phosphorylated and interacts with 14-3-3 in neurons.

We now show that Pavarotti is phosphorylated in neurons at S745 and that this is dependent upon Trc. Please see Figure 3J. To address this, we Immunoprecipitated Pavarotti from control brains of *Drosophila* 3^rd^ instar larvae or brains expressing Trc RNAi in neurons. We carried out western blotting and showed that the phospho signal at S745 is decreased when Trc is depleted.

4) The authors propose in their model that 14-3-3 binding to the phospho-epitope promotes the association of Pavarotti with microtubules and its crosslinking and braking activity. However, Douglas et al., 2010, published that 14-3-3 binding to the S710 residue in human kinesin-6 reduced the MT binding and bundling activity of centralspindlin. The authors should account for this discrepancy in the text.

This discrepancy is now further discussed. Please see the first paragraph of the subsection “Pavarotti requires 14-3-3 proteins to brake microtubule sliding in S2 cells”.

5) Figure 5: The authors show that Pav or Trc knockdown leads to an increase in the total length of dendrites of Class IV neurons. It has been previously reported that Trc knockdown increases branching (Emoto et al., 2004). It is unclear why increased MT sliding would cause an increase in branching as opposed to dendrite growth. Do the authors observe an increase in branching after Pav knockdown? In the images presented, the effect of Pav depletion looks qualitatively different than the effect of Trc depletion; Pavarotti depletion does appear by eye to produce more secondary and tertiary branches than Trc depletion. Is this the case? These parameters should be quantified and discussed.

As the reviewers suggested, we have now quantified number of branch points per neuron in the class IV DA neurons analysed in Figure 5. These data are now presented in Figure 5—figure supplement 1. Interestingly, we find that Trc depletion either alone or in combination with Pav depletion increases number of dendritic branch points (consistent with previous reports). However, depletion of Pavarotti alone gives values similar to those for control cells. These data suggest that branching of the dendritic arbor is independent of microtubule sliding and also suggest that another substrate of Trc might be in charge of controlling the branching process. Therefore, we hypothesize from these data that Trc regulates dendrite outgrowth and dendrite branching via different mechanisms.

6) Figure 5: If possible, these analyses should be carried out at an earlier time point to demonstrate that this phenotype is the result of a defect in initial dendrite outgrowth.

We attempted this analysis at a late embryonic stage when DA neurons can be observed but their development has not progressed very far. Unfortunately, we found that syncing the embryo collection was unreliable and, despite a short time frame for embryo collection (30mins), embryos were clearly in different developmental stages. Thus, the comparison of DA neuron dendrite length would not be accurate between conditions.

7) In Figure 1G (and Video 2), there does not appear to be a clear difference in sliding between the two conditions from the images/video. This should be addressed.

We have now chosen more representative examples. Please refer to Figure 1E and Video 2.

8) There are a number of instances (listed below) throughout the manuscript where knockdown efficiencies and expression levels should be assessed: a) Furry knockdown in Figure 2;

Please see Figure 2—figure supplement 1B and C.

b) Pav depletion and expression experiments in Figure 3C, D;

Please see Figure 3—figure supplement 1. Here we show Pavarotti depletion by non-coding dsRNA by western blot. We also confirm equivalent expression of WT and S745A GFP Pavarotti by immunofluorescence.

c) PavGFP over-expression in Figure 4;

Please Figure 4—figure supplement 1 for demonstration of GFP Pavarotti signal.

d) Knockdown efficiencies of Trc and Pav in the DA neurons in Figure 5. This is a beautiful cell type and assay but it needs to be accompanied by some assessment (IF, WB, RT-qPCR) that the RNAi is working in these cells.

Trc knockdown in neurons is confirmed by western blotting of Trc in control brain lysate and lysate from larvae expressing Trc RNAi under control of the neuron specific driver elav Gal4. Elav gal4 is active in all neurons of the peripheral nervous system from stage 12 of embryo development, therefore will be driving the relevant RNAis in the analysed class IV DA neurons (Luo et al., 1994). Please see Figure 1—figure supplement 1. We also confirm double knockdown of Pavarotti and Trc by elav Gal4 in brain lysate related to Figure 5.

Unfortunately, it has not been possible to confirm knockdown exclusively in class IV DA neurons as these cells represent a very small, and not easily isolatable population, therefore western blotting is not possible. Further, immunofluorescence experiments would include signal from all surrounding cells, occluding any effect seen in these neurons.

9) The measurements of the% microtubule area shown in Figures 3 and 4 are unclear, especially as presented as a single merged image. Please show each individual channel and highlight the area that you designate as "Pav positive". Pav is nuclear in interphase and, while the images are difficult to assess, it appears that the nuclear signal is evident in some of them. How is this taken into account when evaluating the area that is measured?

We now include Figure 3—figure supplement 2 detailing how this analysis was carried out and how the nuclear Pavarotti signal is excluded in this analysis. Each individual channel for the example images is now presented in Figure 4B to clearly demonstrate the Pav positive microtubule area.

10) The authors reference a constitutively-active Trc in the subsection “Trc kinase activity is necessary to control microtubule sliding”, and refer in to data showing that this construct reduces sliding comparably to WT Trc; however, these data are not presented in Figure 2.

We have now expanded Figure 2A and B to include all relevant conditions.

11) The co-IPs in Figure 4A should be quantified. In addition, the experiment shown in 4A is not currently described in sufficient detail so that readers can understand exactly what was done. Please provide a more detailed explanation.

The coIPs in Figure 4A are now quantified as in Figure 3. The experiment is now described in better detail in the Materials and methods section. Please refer to ‘Immunoprecipitation and western blotting’ in this section.

12) Very little is mentioned in the manuscript about the other component of centralspindlin – MgcRacGAP. The authors should clarify if they believe this mechanism is being mediated by the centralspindlin complex. Part of the source of the confusion also stems from the model figure that appears to show a tetramer of Pavarotti – is that the intention or is this supposed to be a Pavarotti dimer and a Tumbleweed dimer? If it is centralspindlin then the complex should be labeled appropriately and the schematic should be changed accordingly (at least different colors for the constituents).

We have now expanded our discussion of MgcRacGAP and included it in the summary figure to show that, while we are focussing on Pavarotti phosphorylation, we believe this mechanism to be carried out by the centralspindlin complex. Please refer to the second paragraph of the subsection “Pavarotti requires 14-3-3 proteins to brake microtubule sliding in S2 cells”, in the Discussion.

[Editors' note: further revisions were suggested prior to acceptance, as described below.]The manuscript has been significantly improved, but there are a few remaining issues that need to be addressed before acceptance, as outlined below:1) Much of the new data relates to extending the findings from *Drosophila* S2 cells further into neurons. To do so the authors have added additional sliding data from cultured neurons to Figure 2 (Figure 2—figure supplement 1) and Figure 5 that agree with the results from S2 cells. They also show that there is a reduction of Pavarotti phosphorylation on S745 following depletion of Trc in fly brains in Figure 3. While the levels of pS745-Pav are quantified, the total Pav levels are not. The authors need to quantify the bands in the lower blot of Figure 3J and report the ratio of pS745 to total Pavarotti.

This is how the western blots have already been quantified throughout all experiments. We apologise for the confusion and have added the following in the Materials and methods section to clarify:

“For quantification of western blots, Licor Image Studio software was used. In each case, phosphorylated Pavarotti levels were compared to total Pavarotti levels. Then this ratio was normalized to that of the control condition. In this way, any difference in immunoprecipitated total Pavarotti levels is accounted for between the two conditions.”

2) The author's claim that 14-3-3 is required for regulation of microtubule sliding is still not explicitly supported by the data presented. Figure 4B and 4C clearly show that there is no effect on sliding when 14-3-3 is inhibited. In light of this, the language in the Abstract should be modified.

The Abstract now reads “Loss of 14-3-3 prevents Pavarotti from associating with microtubules”.

3) "In vitro we show that these two proteins act together to regulate sliding in developing cultured neurons." This sentence needs to be re-written to more clearly indicate that the in vitro studies support a model in which the two proteins act together to regulate sliding in developing cultured neurons.

This line now reads “Our in vitrodata suggest that these two proteins act together to regulate sliding in developing cultured neurons.”